# Motor learning induces myelin-related white matter changes revealed by MRI-based in vivo histology
Norman Aye [1,2], Jörn Kaufmann [3], Hans-Jochen Heinze[3,4,5,6], Emrah Düzel [2,4,5,7,8], Gabriel Ziegler [2,4,7], Marco Taubert [1,2,5] & Nico Lehmann [1,2,9] ✉

Motor learning induces widespread brain changes, yet the microstructural mechanisms underlying human white matter (WM) plasticity remain poorly understood. Animal studies have identified roles for neurites, glia, and myelin, but in vivo human evidence has been limited by measurement specificity. Here, we combine multi-contrast quantitative MRI (qMRI), tractometry, and a novel multivariate analysis framework to investigate the microstructural basis of WM plasticity during motor skill learning. In a longitudinal within-subject study, 24 healthy adults completed 4 weeks of balance training following a baseline control period without training. We mapped changes across tractography-defined WM pathways using complementary qMRI markers related to tissue density, myelin, neurite architecture, and iron. Multivariate analysis revealed biologically plausible, behaviorally relevant plasticity in distributed pathways—including the cortico-ponto-cerebello-thalamo-cortical loop, anterior thalamic radiation, and corticospinal tracts—with important contributions from myelin-related metrics. Notably, we observed changes consistent with training-related modulation of the aggregate *g*-ratio in humans. These spatially distributed effects converged into a single latent dimension predicting neocortical plasticity, suggesting a coordinated, cross-tissue mechanism of brain adaptation. This biologically interpretable framework offers a powerful new approach for investigating WM microstructure in the contexts of plasticity, development, aging, disease, and rehabilitation.

Motor learning is the process by which movement control improves through experience, leading to more dexterous and automatic actions[1]. It is a widely held view that interventions to facilitate motor learning could benefit greatly from a deeper understanding of the underlying neurobiology[2,3].

Motor learning has long been thought to primarily involve changes in synapse number and morphology[4] across a widespread cortical and sub-cortical network[5]. However, recent evidence highlights white matter's (WM) role in synchronizing spike-time arrivals and neural oscillations across distant neuron populations[6–8], underscoring that motor learning extends Hebbian learning mechanisms observable in gray matter (GM).

Ex vivo histological studies in animal models suggest that myelin, glial components, and axons are the primary drivers of experience-dependent WM plasticity[6,9,10]. In humans, the aggregate microstructural changes within

a few cubic millimeters of brain tissue can be indirectly measured with magnetic resonance imaging (MRI)[11]. However, conventional imaging approaches such as T1-weighted MRI or tensor-based representations of diffusion MRI data (DTI) provide only limited biological specificity, as the measured signals cannot be mapped unambiguously to underlying tissue microstructural properties[11,12]. Disentangling the relative contributions of neurites, glial cells, and myelin to WM plasticity would be highly desirable[6], as it could enable more precise monitoring and targeted treatment of conditions such as demyelinating diseases[13] or axonopathy following head trauma[14].

Fortunately, quantitative MRI (qMRI)—that is, advanced MRI pulse sequences combined with biophysically informed modeling of brain tissue[15]—has enabled a more biologically interpretable characterization of

[1]Faculty of Humanities, Institute III, Department of Sport Science, Otto von Guericke University, Magdeburg, Germany. [2]Collaborative Research Center 1436 Neural Resources of Cognition, Otto von Guericke University, Magdeburg, Germany. [3]Department of Neurology, Otto von Guericke University, Magdeburg, Germany. [4]German Center for Neurodegenerative Diseases (DZNE), Magdeburg, Germany. [5]Center for Behavioral and Brain Science (CBBS), Otto von Guericke University, Magdeburg, Germany. [6]Leibniz-Institute for Neurobiology (LIN), Magdeburg, Germany. [7]Institute of Cognitive Neurology and Dementia Research, Otto von Guericke University, Magdeburg, Germany. [8]Institute of Cognitive Neuroscience, University College London, London, UK. [9]Department of Neurology, Max Planck Institute for Human Cognitive and Brain Sciences, Leipzig, Germany. ✉e-mail: nico1.lehmann@ovgu.de

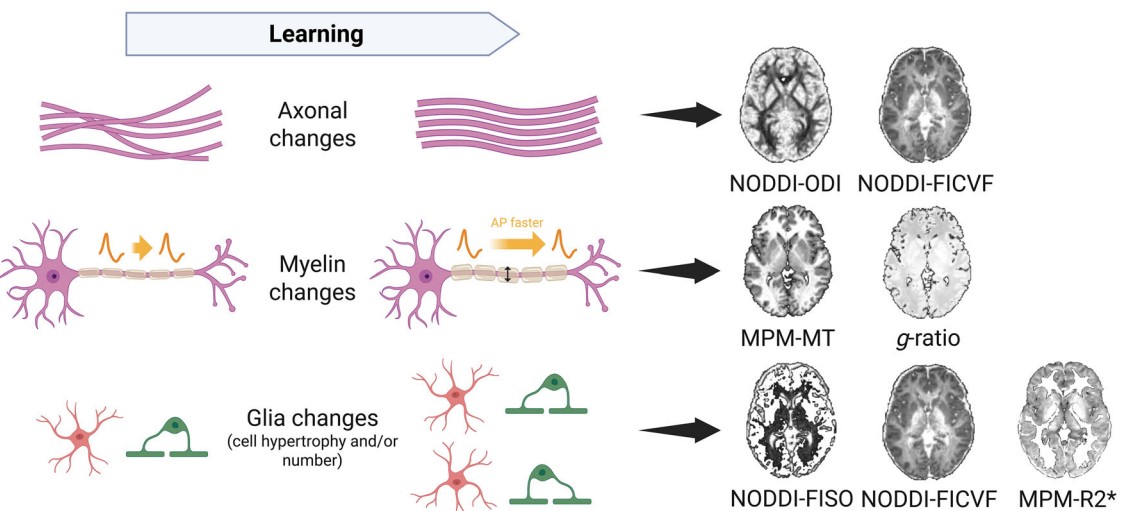

**Fig. 1 | Schematic overview of proposed mechanisms of structural neuroplasticity in white matter (WM), as inferred from animal studies, and the quantitative MRI (qMRI) metrics that have been reported to be associated with these microstructural features (based on refs. 6,8,13,17,22,26,54,97).** From top to bottom, the illustrated cellular and microstructural mechanisms include changes in axonal diameter and mesoscale neurite organization, myelination, and glial cell number and/or hypertrophy. The schematic depicts associations between these processes and MRI-derived measures without implying a specific direction of change. Although several additional MRI metrics (including some illustrated) are sensitive to alterations in axonal properties, myelin, and neuroglia, only those with the highest relative specificity are shown here to maintain conceptual clarity. As a schematic abstraction, the figure does not capture the full, multifaceted nature of myelin plasticity, which may involve changes in sheath thickness, internode length, node of Ranvier properties, or oligodendrocyte progenitor dynamics[6,8,97]. Created in BioRender. Lehmann, N. (2026) https://BioRender.com/33lse6o.

microstructural brain changes[14,16]. For example, fitting the biophysical model Neurite Orientation Dispersion and Density Imaging (NODDI[17]) to high-gradient diffusion MRI data provides estimates of local tissue properties, including the isotropic volume fraction (FISO), the intracellular volume fraction (FICVF), and the orientational coherence of the neurites (orientation dispersion index, ODI). Although these parameters cannot be interpreted as direct histological measurements, they have been shown to correlate with microstructural tissue properties ex vivo[18–21]. Compared with conventional DTI, they provide improved specificity and serve as sensitive markers for detecting microstructural changes[22]. In a similar vein, a semi-quantitative imaging protocol (multi-parameter mapping, MPM) can be utilized to measure longitudinal (R1) and effective transverse relaxation rates (R2*), proton density (PD), and magnetization transfer saturation (MT)[23,24], providing complementary contrasts that relate to distinct, though not unique, biological tissue features such as myelin, iron, and water content (Fig. 1).

Despite these promising advancements, the complete potential of qMRI can only be realized through the effective combination of multiple contrasts[25]. For example, combining myelin-sensitive and diffusion imaging enables the calculation of the aggregate g-ratio[26], a biophysically informed metric interpretable in terms of relative axonal myelination[13]. Furthermore, multivariate feature extraction exploits shared contrast sensitivities to unveil microstructural information that is concealed when considering single modalities in isolation[25].

This study set out to shine new light on learning-induced neuroplasticity by capitalizing on recent advances in qMRI and multivariate statistics. We investigate neuroplasticity after four weeks of learning a dynamic balancing task (DBT;[27,]) which is known to induce plasticity in the WM[27,28] (Fig. 2). We use diffusion tractography to delineate a prespecified set of WM fiber tracts that connect hub regions of the motor network, before multi-contrast qMRI maps reflecting different aspects of tissue were projected to these bundles. Finally, latent microstructural changes over time along the tract trajectories were analyzed[29] and related to concomitant changes in DBT performance. To our knowledge, this is one of the first studies to investigate learning-induced neuroplasticity using multivariate analysis of multi-contrast qMRI, providing a biologically meaningful view of WM changes driven by learning[11, 25].

## Results

We assessed longitudinal microstructural changes along motor-related white matter tracts using NODDI[17] and MPM[23,24] metrics projected onto tractography-defined bundles. Bundle-specific, along-tract statistics[30–32] and longitudinal multivariate analysis[29] enabled spatially resolved, biologically informed insights into white matter plasticity and its behavioral relevance.

### Motor learning

Motor learning was quantified by fitting each subject's DBT performance across eight training sessions with a generalized power function[33]. The resulting exponent was significantly greater than zero, $t(23) = 13.84$, $p < 0.001$, indicating that subjects showed robust improvement in task performance over time. Notably, every participant improved on the task without exception[33].

### Microstructural plasticity in white matter

To infer training-related white matter plasticity, we applied Repeated-measures analysis of variance simultaneous component analysis (RM-ASCA[+ 29]) to capture multivariate longitudinal changes in microstructural metrics across tract segments. Segments showing significant changes in the first principal component (PC1) derived from RM-ASCA[+] and correlating with individual motor learning rates were retained. For interpretability, neighboring segments with similar factor structures and loadings were averaged to emphasize spatially consistent effects.

Five tract segments met the criteria for behaviorally relevant microstructural plasticity. Results are shown in Fig. 3, with each row displaying the RM-ASCA[+] results for a significant segment of a given fiber tract. In all cases, PC1 differed significantly between MRI3 and both pre-training time points (MRI1 and MRI2), as indicated by non-overlapping 95% bootstrap confidence intervals[29]. In each case, no significant changes in PC1 scores were observed during the control period without training (i.e., overlapping CIs), underscoring the reliability of the measurements[34,35].

We now proceed with a detailed presentation of the results shown in Fig. 3. The thalamus is known for its reciprocal connections with the cerebral cortex. One key thalamocortical pathway, the anterior thalamic radiation (ATR), transmits information between the thalamus and the prefrontal cortex[36]. In the right ATR at tract segments 65–67, RM-ASCA[+]

**Fig. 2 | Study design overview.** Participants underwent three MRI sessions, with two four-week intervals: a no-intervention control period (MRI1–MRI2) followed by a motor learning period (MRI2–MRI3) involving eight dynamic balance training sessions (two per week).

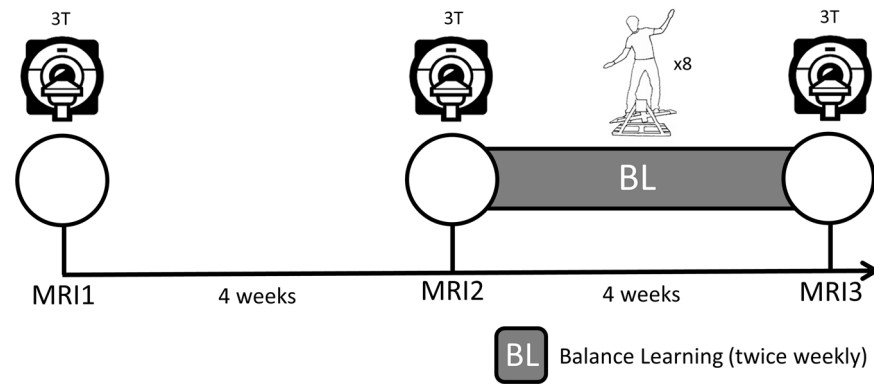

analysis revealed that PC1 explains ≈ 81% of the variance in the time effect. Examination of PC1's trajectory revealed stable factor scores during the control period, as indicated by overlapping confidence intervals between MRI1, 95% BCa CI [-0.07, 0.56], and MRI2, 95% BCa CI [-0.08, 0.52]. In contrast, a significant decrease was observed at MRI3, 95% BCa CI [-0.81, -0.17], evidenced by non-overlapping CIs compared to both earlier time points. The loadings plot indicates that free water fraction (from NODDI) strongly correlates with the trajectory of PC1, suggesting that the decrease in PC1 at MRI3 is primarily driven by a reduction in free water — or, equivalently, an increase in local tissue volume[17,22].

Three fiber bundles demonstrated factor loadings suggestive of possible myelin-related changes. This applied to the right fronto-pontine fibers (FPT) at segment 50 (MRI1, 95% BCa CI [-0.01, 0.61]; MRI2, 95% BCa CI [-0.18, 0.46]; MRI3, 95% BCa CI [-0.76, -0.19]), which project from the frontal lobe to the pontine nuclei and then via the middle cerebellar peduncle to the contralateral cerebellum (PC1 explains ≈ 77% of the variance in the time effect); the thalamic projection to the right premotor cortex (T_PREM) at segments 73–74 (MRI1, 95% BCa CI [0.08, 0.69]; MRI2, 95% BCa CI [-0.21, 0.48]; MRI3, 95% BCa CI [-0.80, -0.24]), part of the dentato-rubro-thalamic-cortical system—an important efferent pathway of the cerebellum[37, 38]—where PC1 explains ≈ 90% of the variance in the time effect; and the right corticospinal tract (CST) at segment 53 (MRI1, 95% BCa CI [0, 0.70]; MRI2, 95% BCa CI [-0.13, 0.58]; MRI3, 95% BCa CI [-1.01, -0.26]), formed by monosynaptic connections between primary motor cortex neurons and motoneurons in the spinal cord (PC1 explains ≈79% of the variance in the time effect). In each of these structures (T_PREM_right, FPT_right, CST_right), a decrease in the aggregate $g$-ratio significantly influenced the trajectory of PC1. Occasional significant loadings were also observed for MT, R1, FISO, and FICVF.

In an inferior section of the left CST at segments 88–89 (MRI1, 95% BCa CI [–0.38, 0.24]; MRI2, 95% BCa CI [–0.68, –0.12]; MRI3, 95% BCa CI [0.26, 0.79]), the latent change score (PC1 explains ≈ 74% of the variance in the time effect) was primarily driven by a concomitant increase in R2* and PD.

In an additional sensitivity analysis, female participants ($n = 3$) were excluded, and the analyses were repeated in the remaining male participants. The results were largely consistent with the primary analyses, with some outcome measures showing slightly increased stability during the control period and/or more pronounced neuroplastic effects following training. Importantly, no qualitative changes in the pattern of results were observed. These findings suggest that inclusion of the female participants does not materially affect the robustness of our conclusions (Supplementary Fig. 1).

No significant latent microstructural changes were found in the other TOIs.

## Longitudinal correlated brain–behavior changes
To infer behaviorally relevant neuroplasticity, microstructural changes in response to training must also relate to individual differences motor learning[39]. In all five tract sections, aggregated correlations between

microstructural change and DBT learning rate across the MRI1–MRI3 and MRI2–MRI3 time intervals were statistically significant (Fig. 4). The mean effect sizes and corresponding 95% CIs[40,41] were as follows: ATR (RH): $G(r) = -0.38$, 95% CI [-0.63, -0.13], $p = .001$, T_PREM (RH): $G(r) = -0.31$, 95% CI [-0.57, -0.05], $p = .01$, FPT (RH): $G(r) = -0.31$, 95% CI [-0.57, -0.05], $p = .01$, CST (RH): $G(r) = -0.30$, 95% CI [-0.57, -0.04], $p = .01$, and CST (LH): $G(r) = 0.35$, 95% CI [0.09, 0.60], $p = .003$.

## Robustness check
Across 2,320 comparisons (29 tracts × 80 segments each), nine tract segments exhibited learning-related microstructural changes that correlated with behavioral improvement (ATR_R: 3 neighboring segments; T_PREM: 2; CST_L: 2; CST_R: 1; FPT_R: 1). Monte Carlo simulations indicated that the probability of observing nine or more significant segments under the null hypothesis was low (Monto Carlo $p = 0.01$; Fig. 5), confirming that these patterns are unlikely to arise by chance.

## Coordinated white matter and cross-tissue plasticity
Finally, we examined whether latent microstructural changes in WM were interrelated and might reflect a shared, motor learning-relevant network. To this end, we subjected the five latent microstructural change scores (PC1's derived from RM-ASCA$^+$) to an additional PCA (unrotated axes). The first principal component accounted for approximately 62% of the variance, suggesting that learning-induced microstructural changes across distributed WM tracts largely form a unidimensional construct (Table 1). This is notable given that the PC1s of the individual tract segments were driven by distinct sets of contributing variables (Fig. 3).

Experience-dependent synaptic plasticity in gray matter is fundamentally linked to the optimal conduction velocity of individual axons, enabling the precisely timed arrival of action potentials at key relay points within the network[7,8,42]. Concurrently, the essential supporting role of white matter neuroglia in modulating plasticity is increasingly recognized[6,9]. Based on these converging insights, we investigated cross-tissue plasticity by modeling previously reported training-related changes in cortical neurite orientation dispersion (ODI) in the same subjects[33] as a function of the distributed white matter microstructural changes summarized by the principal component (Table 1), using robust simple regression. This analysis yielded a significant result, $b = -1.82$, 95% BCa CI [–3.19, –0.55], $p = 0.005$, $R^2 = 0.19$ (Fig. 6). Similarly, a multiple regression model using the five PC1 scores (derived from RM-ASCA$^+$) as individual predictors of cortical neurite orientation changes also showed a significant model fit, multiple $R^2 = 0.36$, $p = 0.046$. Due to the high degree of multicollinearity among predictors (cf. Table 1), we do not report individual regression coefficients. The difference in explanatory power between the two models (19% vs. 36% of variance explained) is largely attributable to additional criterion-relevant variance contributed by the two CST tract sections (see factor structure, Table 1). Leave-One-Out Cross-Validation (LOOCV) results suggest that while both robust regression models capture meaningful relationships in this sample, their predictive generalizability to independent data may be limited (Table 2).

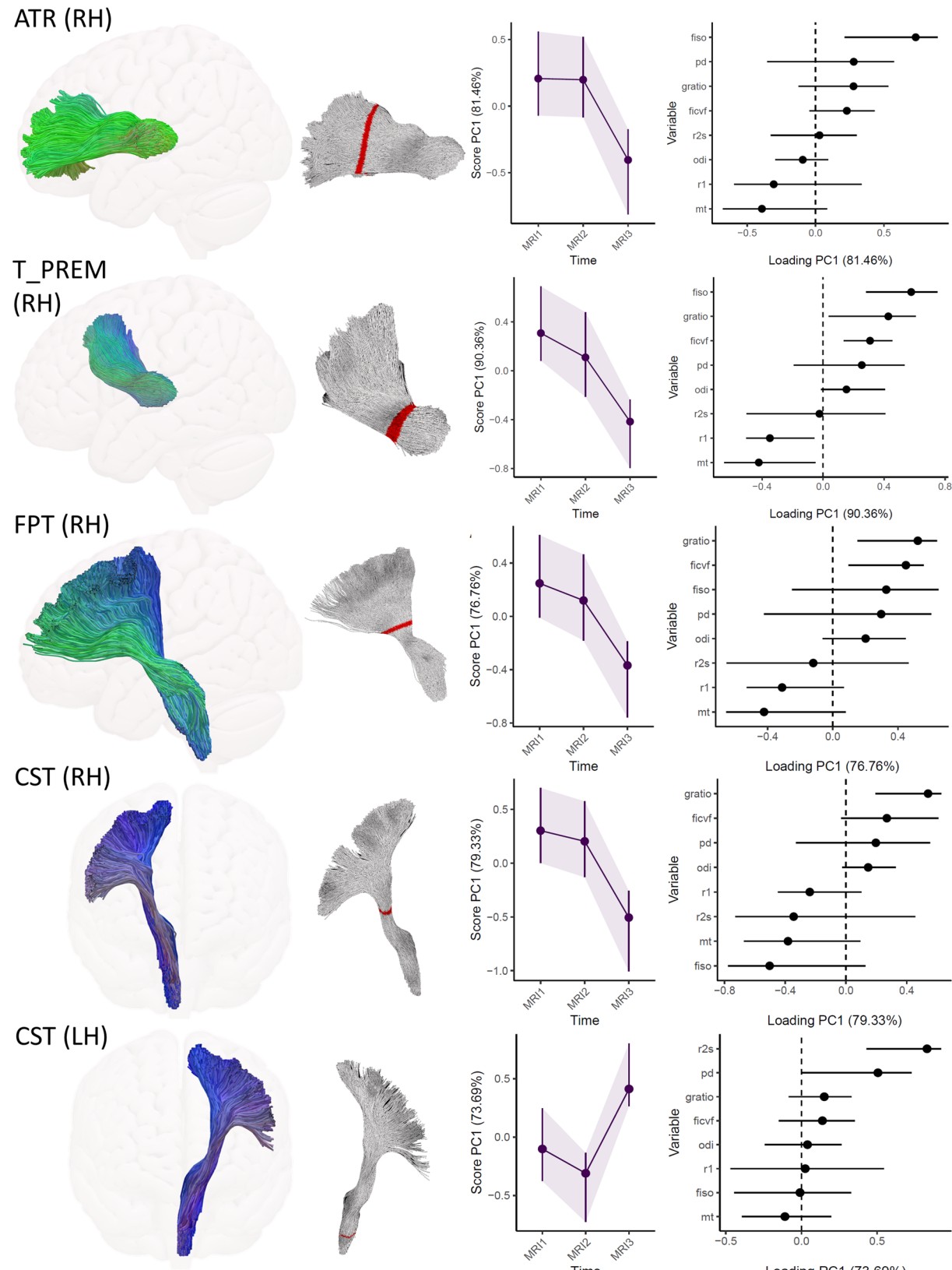

## Discussion

Progress toward a deeper understanding of structural brain changes during motor learning has been hindered by two key challenges. First, many imaging techniques are of limited specificity to the underlying biological processes[11]. Second, there has been a lack of comprehensive analytical strategies capable of integrating insights from multiple, complementary MRI-derived metrics—particularly in longitudinal designs[25]. These methodological constraints have resulted in ambiguity regarding the relative contributions of various tissue components, such as neurites, glia, and myelin, to the signal alterations detectable through conventional MRI[6,11]. In

**Fig. 3 | Microstructural plasticity in response to motor learning in young adults in five white matter fiber tracts.** *Left column.* 3D representations of the virtually dissected tracts. *Middle left column.* Locations of tract segments showing significant multivariate change patterns, as described below. *Middle right column.* Multivariate microstructural trajectories based on a PCA on the linear mixed model (LMM) effect matrices for the time effect. RM-ASCA+ extracted a general change pattern represented by PC1, with the proportion of explained variance indicated on the y-axis. Note that in all five tracts, the bootstrapped 95% CIs for MRI3 do not overlap with those for the control period (MRI1, MRI2), indicating significant learning-induced change[29]. *Right column.* Loading plots. Positive loadings on PC1 indicate that a microstructural metric follows the trajectory of PC1; negative loadings indicate that increases (or decreases) in PC1 are accompanied by decreases (or increases) in that metric. Error bars represent 95% CIs, with non-overlapping intervals relative to zero indicating statistically significant loadings. Larger absolute loading values reflect a stronger influence of that metric on PC1. ATR anterior thalamic radiation, T_PREM thalamopremotor tract, FPT frontopontine tract, CST corticospinal tract, LH left hemisphere, RH right hemisphere.

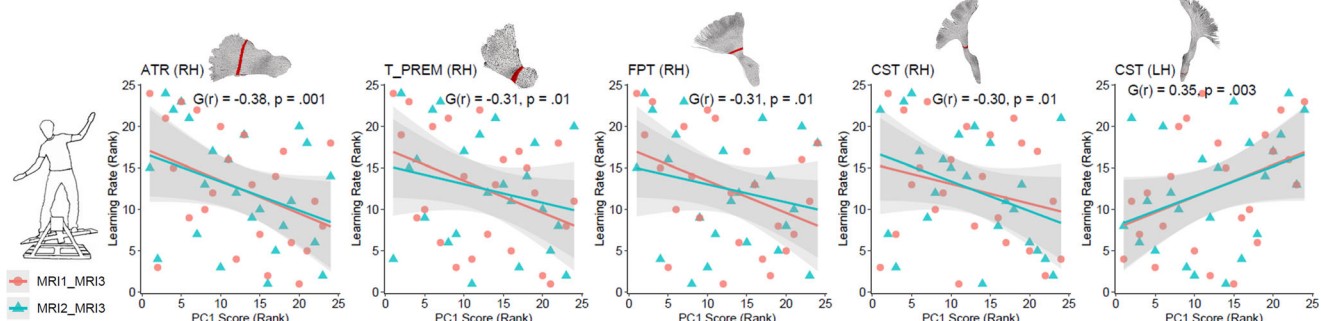

**Fig. 4 | Rank correlations between learning-induced latent microstructural changes—across two time intervals (MRI1–MRI3 and MRI2–MRI3)—and DBT learning rate.** Results reflect the expected direction of effects: PCs that decreased following DBT training (Fig. 3) show negative correlations with balance learning, and vice versa. Annotated pooled correlation coefficients were derived from fixed-effect meta-analyses[40,41].

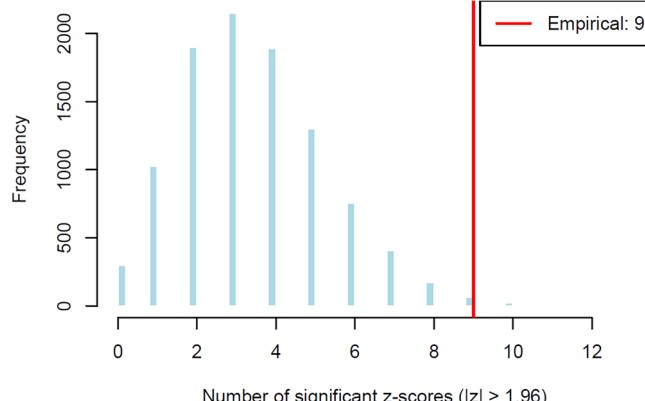

**Fig. 5 | Histogram of the simulated distribution of significant z-scores (|z| > 1.96) under the multivariate null hypothesis, based on 10,000 Monte Carlo simulations.** In each simulation, 2320 sets of four correlated z-scores were generated ($\rho = 0.5$ across z-scores). For each set, the z-score with the smallest absolute value—that is, the least significant one—was selected. The number of selected z-scores exceeding the significance threshold was then counted. The red vertical line marks the empirically observed number of significant results ($n = 9$).

the present study, we attempted to investigate the structural underpinnings of motor learning in humans by combining multi-contrast qMRI with a multivariate analysis approach focusing on latent changes in brain microstructure. At a localized level, we registered behaviorally relevant microstructural changes in segments of the right cortico-ponto-cerebello-thalamo-cortical loop, the right anterior thalamic radiation, and the bilateral corticospinal tracts — fiber tracts known to interconnect brain areas crucial to motor control and learning[5]. Consistent with theoretical predictions[6,8,10], the observed patterns are suggestive of myelin-related changes and broader alterations in tissue integrity underlying white matter plasticity. Importantly, these distributed WM changes were not random, and their convergence into a single latent variable may reflect a coordinated, network-wide response to training.

Voluntary movements depend on many areas of the brain, which collaborate to accomplish subtasks such as sensory processing, motor planning, action selection, execution of precisely timed efferents, and error correction[5]. Studies based on functional and structural MRI have consistently shown that the thalamus, an often neglected but highly important connector and processing hub of the brain[38], plays a crucial role in balance control and learning[43,44]. Accordingly, it is unsurprising that thalamic fiber tracts showed microstructural alterations following training, with these changes correlating with motor learning performance. On one hand, this applied to the descending (fronto-pontine) and ascending (thalamo-pre-motor) pathways of the cortico-ponto-cerebello-thalamo-cortical circuit, which connects the cortex to the contralateral cerebellum via the pons and projects back via the thalamus[37,38]. On the other hand, we observed microstructural changes in the anterior thalamic radiation, reciprocally connecting the thalamus and the prefrontal cortex[36]. Taken together, these changes likely reflect how motor learning strains thalamus-dependent processes – relaying and integrating sensory-motor information and supporting complex movement planning[36,45–47]. Behaviorally relevant microstructural changes were also detected in the bilateral corticospinal tracts, which synapse directly onto spinal motoneurons to support fine voluntary motor control[48].

While the occurrence of WM plasticity in thalamic fiber tracts and the CST agrees well with previous training studies using the DBT paradigm[27,49], the present study sheds new light on the microstructural processes that were altered by training. The aggregate *g*-ratio, a biophysically informed metric developed to index relative axonal myelination[13,26], decreased following training in three tracts (FPT_R, T_PREM_R, CST_R) and ranked among the top contributors to latent microstructural changes. Although speculative, this change might help synchronize neural activity across distributed brain regions involved in movement[8], consistent with findings on thalamocortical motor projections in animal models[42]. The strength of the multivariate approach is particularly evident in interpreting behaviorally relevant plasticity in the right anterior thalamic radiation (ATR_R). Here, a decrease in the free water fraction might indicate enhanced tissue integrity and/or reduced extracellular space[17,22]. This pattern could also, in theory, reflect glia-related adaptations[17,50], since the tendency towards decreased intracellular volume fraction may be consistent with an expansion of

## Table 1 | Factor loadings of the original variables (PC1s from RM-ASCA$^+$) on the first principal component (PC1) extracted from the data via PCA

| | ATR (RH) | T_PREM (RH) | FPT (RH) | CST (RH) | CST (LH) |
|---|---|---|---|---|---|
| loading on PC1 (62%) | 0.87 | 0.88 | 0.91 | 0.69 | -0.49 |

Parallel analysis[96] supported the retention of a single factor. Assumptions for PCA were adequately met[90,91]: the overall Kaiser-Meyer-Olkin (KMO) measure was 0.78 (individual KMOs: 0.74–0.83), indicating good sampling adequacy; Bartlett's test of sphericity was significant, $\chi^2(10) = 55.73$, $p < .001$, confirming sufficient intercorrelations among variables; and the determinant of the correlation matrix exceeded 0.00001[90], suggesting multicollinearity was not a concern.

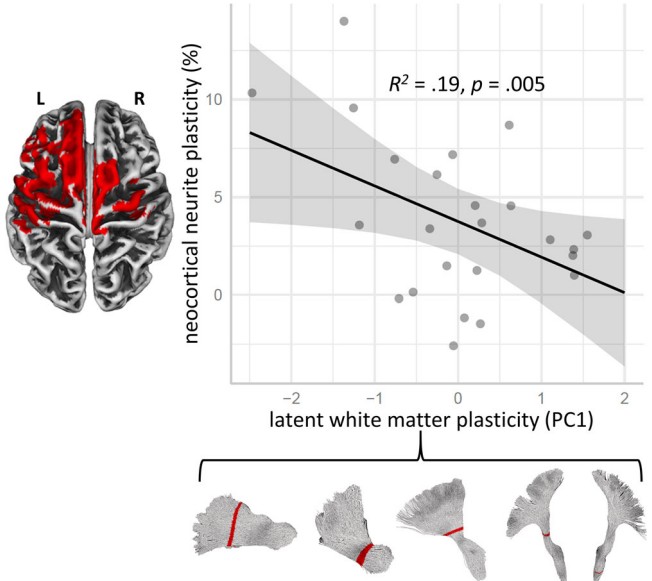

**Fig. 6 | Negative relationship between latent white matter plasticity (PC1 score from Table 1) and post-training changes in neocortical neurite orientation dispersion index (ODI; Ref. 33) estimated using robust simple regression[95].** The gray band represents the 95% confidence interval of the regression line. Below, tract reconstructions indicate the five white matter segments contributing to the latent variable. The cortical surface (left) highlights regions showing significant post-training ODI increases (red; $p < 0.05$, corrected[33]).

neuroglia[17,22]; however, this interpretation remains tentative given wide confidence intervals. Notably, glial cells are thought to influence white matter MR signals[10], and the absence of significant changes in myelin-related metrics is compatible with this explanation. While oligodendrocyte-driven myelination is well established[6,51], astrocytic roles in metabolic support, waste clearance, and network modulation are increasingly recognized[9]. Finally, in an inferior segment of the left CST, increases in both PD and R2* were observed to load onto a shared latent change component. While these changes correlated with motor learning, their biological interpretation remains uncertain. An increase in PD could be consistent with transient cellular phenomena, such as astrocytic swelling[52,53]. The concurrent R2* increase might be related to iron-associated processes—either glial changes involving iron deposits in neuroglial somata[54] or vascular alterations linked to heme-bound iron[55]. Alternatively, the rise in R2* could reflect the influence of PD itself, since elevated water content can enhance magnetic susceptibility and accelerate relaxation. However, given the at best moderate association between PD and R2* changes in this tract segment ($r \leq 0.33$, n.s.), any such interdependence appears limited.

Motor functions arise from coordinated information processing across distributed yet interconnected neural networks[5]. One feature thought to reflect such shared network organization is structural covariance[56]; accordingly, correlated microstructural changes across regions may indicate reorganization within an underlying network. In line with this, PCA

revealed intercorrelated microstructural plasticity across distinct white matter tracts, even though the specific biological mechanisms driving these changes differed between tracts. Notably, these distributed white matter alterations also showed associations with neocortical plasticity in the same sample[33], suggesting a potential coordinated, network-level adaptation. However, LOOCV analyses indicate that these associations may be specific to this sample.

Some methodological aspects warrant discussion. First, PCA was applied to each tract segment individually, rather than to entire bundles or across bundles as in prior studies[57,58]. This segment-level approach is justified because training-induced white-matter changes are often spatially localized, and microstructure is heterogeneous even within a single tract[59], so interactions among features can differ along and between tracts. Second, tract-specific interpretation may be limited by partial overlap of significant FPT and T_PREM segments within the internal capsule, a classical white-matter "bottleneck" region[60]. Segment-level averaging, while potentially diluting highly localized effects, represents a practical compromise: it enhances power and reduces noise relative to voxel-wise analyses while retaining anatomical specificity[30–32,50,59]. Pure voxel-wise approaches offer higher spatial resolution but face challenges such as misregistration, low signal-to-noise ratio, and severe multiple-comparison penalties[34,61]. Third, in RM-ASCA$^+$, conventional $p$-value–based significance testing is not currently possible. We therefore assessed robustness using bootstrapped confidence intervals for scores and loadings[29], and addressed multiple-testing concerns with Monte Carlo simulations; alternative approaches may emerge in the future. Fourth, the aggregate $g$-ratio map was anchored to a splenial reference value of 0.70[62,63]. While consistent with prior studies and supported by histology[64], the true in vivo $g$-ratio may vary across white-matter regions[65]. Sensitivity analyses with reference values between 0.68 and 0.72 yielded near-perfect correlations with the original maps, supporting robustness within this physiologically plausible range.

Beyond these, additional limitations relate to sample characteristics and statistical power. The moderate sample size limits sensitivity to detect tract-specific effects of modest magnitude. Balance training likely induces broadly distributed white-matter adaptations rather than focal changes[44], but stringent plasticity criteria as applied here reduce the probability of detecting modest tract-specific effects in any single tract. In addition, the sex distribution was unbalanced, with only three female participants, so the findings predominantly reflect males.

In summary, this study provides new insights into neuroplasticity by introducing a principled approach that combines multivariate analysis of reliable[34,35] multi-contrast qMRI metrics within a diffusion tractometry framework. Consistent with theoretical predictions and prior human studies, our findings are consistent with myelin-related changes and broader alterations in general tissue integrity as potential contributors to white matter plasticity[50,66,67]. A strong case can be made for the validity of these results; the neuroplasticity observed in different WM tracts was confined to the training phase and was behaviorally relevant. Notably, the latent WM changes among distant tracts were strongly interrelated, bridging the relationship between microstructural tissue features and network-level reorganization. These changes also correlated with concomitant neocortical microstructural alterations in motor-relevant brain areas that were investigated independently[33]. To better understand how neural changes unfold as individuals acquire a new motor skill, future research should consider employing a greater number of MRI measurements. We hope these findings will inspire further research into the brain mechanisms that can be leveraged to enhance plasticity and, in turn, improve motor learning in both healthy individuals and those with clinical conditions. Our approach also offers a powerful tool for tracking responses to training and therapy, opening new avenues in both basic and translational neuroscience.

## Methods
### Participants and experimental design
We used a controlled within-subject design to test whether learning the DBT alters WM microstructure. Twenty-four healthy, right-handed adults

**Table 2 | Leave-One-Out Cross-Validation (LOOCV) of robust regression models**

| model | no. of predictors | $R^2$ (original model) | LOOCV MSE | LOOCV RMSE | 1 - (SSE_model ÷ SSE_null) |
|---|---|---|---|---|---|
| Version 1 | 1 | .19 | 22.36 | 4.73 | 0.07 |
| Version 2 | 5 | .36 | 26.46 | 5.14 | -0.10 |

Model accuracy was quantified using mean squared error (MSE), root mean squared error (RMSE), and the ratio of the model's sum of squared errors (SSE_model) to a null model always predicting the training mean (SSE_null). Positive ratios indicate predictive power; negative ratios indicate overfitting.

(3 women; age: $M = 22.21$, $SD = 3.05$, range 19–29; BMI: $M = 23.55$, $SD = 2.52$, range 18.99–29.40) with no history of neurological, psychiatric, or systemic illness participated.

Exclusion criteria included MRI contraindications, BMI > 30 kg/m², high physical activity (>2 h/week), prior experience with the DBT, and past or current performance-oriented participation in endurance, balance-, or coordination-intensive sports. Participants were drawn from earlier qMRI reliability studies[34,35]. The study was approved by the Ethics Committee of Otto von Guericke University Magdeburg (106/98), and all participants gave written informed consent. All ethical regulations relevant to human research participants were followed.

The experiment included a four-week control phase and a four-week DBT training phase (Fig. 2), with MRI scans at three time points (spaced four weeks apart). The first interval (MRI1–MRI2) assessed qMRI reliability[34,35]; the second (MRI2–MRI3) captured training-related plasticity. To avoid acute training effects, MRI scans were conducted at least 24 h apart from any DBT training session.

### Dynamic whole body balancing task (DBT)

Following the second MRI session (MRI2), participants underwent eight training sessions of a dynamic balancing task (DBT) using a seesaw-like stability platform (Model 16030, Lafayette Instruments, Lafayette, IN, USA). The platform permits medio-lateral movement with a maximum tilt of ±26° on each side. Training was conducted over a four-week intervention period, with participants completing two sessions per week, spaced at least 24 hours apart.

Each session comprised 15 trials, each lasting 30 seconds. Participants were instructed to maintain the platform within ±3° of horizontal for as long as possible during each trial, while fixating on a cross placed at eye level on the wall in front of them[27]. To prevent fatigue, a 2-minute rest interval was provided between trials.

The primary outcome was the balance time (in seconds), defined as the cumulative duration per trial during which the platform remained within the target range of ±3°. After each trial, participants received verbal feedback on their balance time, but no guidance on movement strategies was provided, consistent with a discovery learning approach.

To quantify individual learning trajectories, performance data from all training sessions and trials were modeled using a general power function fitted to within-session DBT averages over time[28,49].

### MRI image acquisition

MRI data were acquired on a 3T MAGNETOM Prisma scanner (Siemens Healthcare, Erlangen, Germany) using a 64-channel head coil. The same imaging protocol was applied consistently across all participants and scanning sessions. To minimize head motion and ensure consistent positioning both within and between subjects, foam padding was placed around the sides and back of the head. Participants were instructed to relax, clear their minds, and remain as still as possible during scanning.

### Diffusion imaging data

Whole-brain diffusion-weighted images were acquired using a monopolar single-shot spin-echo echo planar imaging (EPI) sequence with the following parameters: TE = 74 ms, TR = 4970 ms, flip angle = 90°, GRAPPA acceleration factor = 2, matrix size = 130 × 130, field of view (FOV) = 208 × 208 mm², nominal voxel size = 1.6 × 1.6 × 1.6 mm³, and multiband acceleration factor = 2. The phase-encoding direction was anterior-to-posterior.

The diffusion sampling scheme was based on Caruyer et al.[68], comprising 228 isotropically distributed diffusion-weighted directions across three shells with b-values of 1000 s/mm² (38 directions), 2000 s/mm² (76 directions), and 3000 s/mm² (114 directions). This acquisition provides sufficient diffusion weighting to probe water pools exhibiting restricted diffusion (e.g., within neurites) while reducing signal contributions from faster diffusing extracellular water compartments[12,15]. Fourteen non-diffusion-weighted ($b = 0$ s/mm²) images were interspersed throughout the scan. To correct for susceptibility-induced distortions, nine additional $b = 0$ images were acquired with reversed phase-encoding (posterior-to-anterior).

The total scan time for the diffusion protocol was 22 minutes and 31 seconds.

### Multiparameter mapping

The Multi-Parameter Mapping (MPM) protocol[23,24] provides high-resolution, (semi)quantitative maps of complementary tissue properties, including magnetization transfer saturation (MT, sensitive to macromolecular content such as myelin), proton density (PD, reflecting tissue water content), transverse relaxation rate (R2*, sensitive to magnetic susceptibility, primarily driven by iron), and effective longitudinal relaxation rate (R1, sensitive—but not specific—to myelin, iron, and macromolecular content)[15,54].

MPMs were acquired using three multi-echo 3D fast low-angle shot (FLASH) scans, each predominantly T1-, PD-, or MT-weighted[24], with optimized repetition times (TR) and flip angles (α): TR/α = 23.0 ms/25° for T1-weighted, 23.0 ms/5° for PD-weighted, and 37.0 ms/7° for MT-weighted acquisitions. T1w and PDw scans included 8 echoes at equidistant echo times (2.46–19.68 ms), MTw scans included 6 echoes (2.46–14.76 ms), all with alternating readout polarity. Additional parameters: 0.8 mm isotropic resolution, 224 sagittal slices, FOV = 230 × 230 mm.

Flip angles were optimized using a semi-empirical approach[69] to balance signal-to-noise ratio (SNR) and minimize RF spoiling bias. For T1w and PDw scans, angles were scaled from the Ernst angle by factors of 2.4142 and 0.4142, respectively[70]. To reduce transverse coherence effects at higher flip angles[69], the T1w angle was empirically lowered to 25°[71].

Transmit and receive field calibration scans were performed prior to each weighted scan: B1- mapping: 56 sagittal slices, TR = 4.1 ms, TE = 1.98 ms, FOV = 230 × 230 mm; B1+ mapping: 24 sagittal slices (5 mm thickness), TR = 2,000 ms, TE1 = TE2 = 14 ms, flip angles = 90°, 120°, 60°, 135°, 45°[72,73].

The total acquisition time for the full MPM protocol was 34 minutes and 23 seconds.

### Spatial processing

**Diffusion-weighted images: preprocessing and map fitting.** Preprocessing of diffusion-weighted images followed the standard FSL diffusion pipeline[74]. Following a comprehensive visual quality assessment, data were corrected for susceptibility-induced distortions using the blip-up/blip-down method implemented in FSL's *topup*[75], and for eddy current-induced distortions and subject motion using *eddy*[76]. Image realignment during motion correction was accompanied by appropriate rotation of the diffusion gradient vectors[77].

Microstructural parameter maps were estimated using the NODDI model[17], applied to the preprocessed multishell diffusion data ($b = 0, 1000, 2000,$ and 3000 s/mm²) using the NODDI Matlab Toolbox v1.0.1 (http://

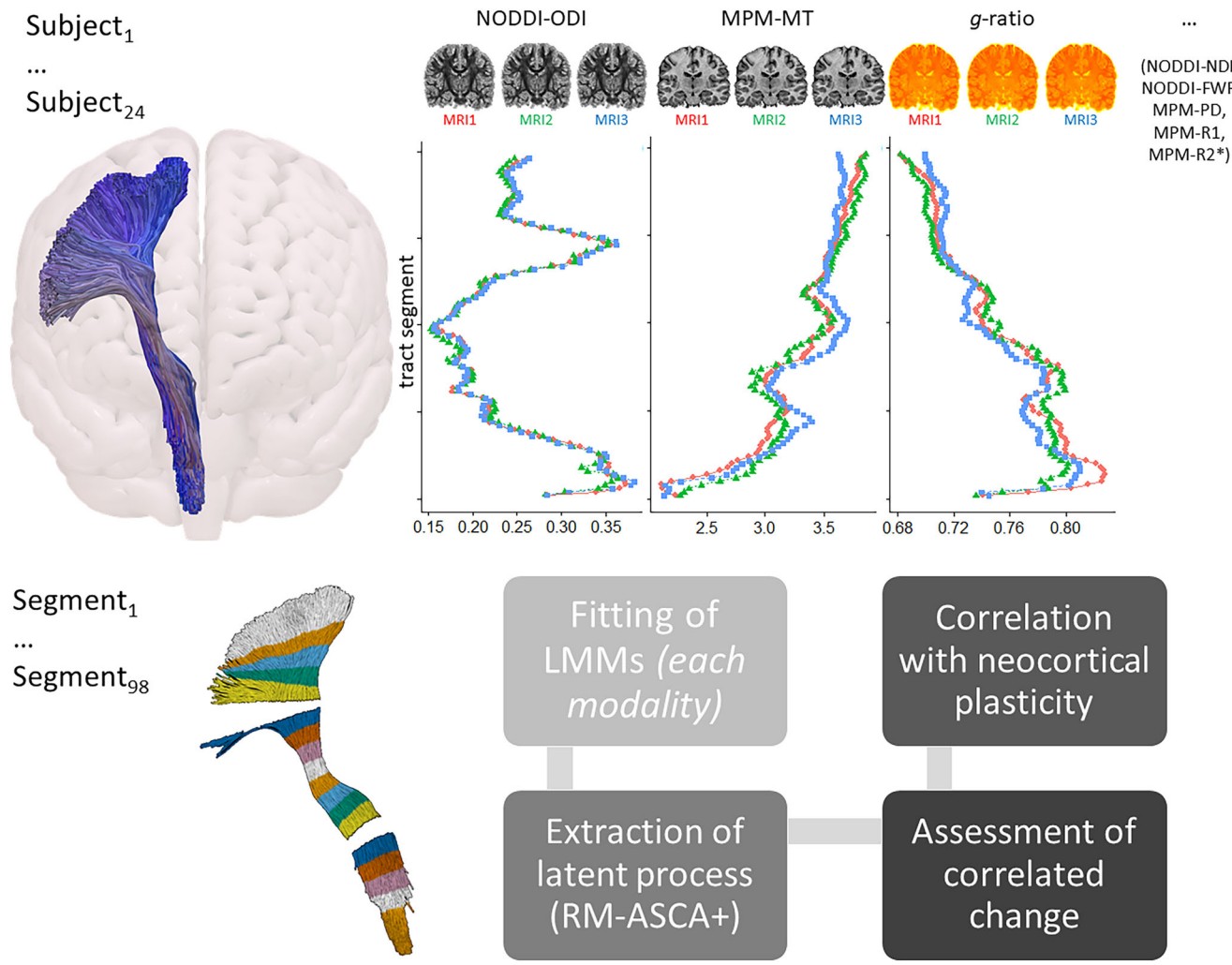

**Fig. 7 | Schematic overview of the analysis approach.** Upper panel, For each of the $n = 24$ subjects, three MRI measurements were acquired four weeks apart. Diffusion tractography was used to segment tracts of interest, and multiple microstructural tissue metrics were projected onto the segmented tracts (here: CST). In addition to the metrics shown here—Orientation Dispersion Index (ODI; NODDI), Magnetization Transfer (MT; MPM), and the aggregate g-ratio—seven other imaging metrics were analyzed (see main text for details). Previous tractometry studies typically performed univariate tract profiling[59], i.e., a single metric was analyzed across the length of the tract (for example, along 98 tract segments). **Lower panel**, In contrast to this, our approach involves longitudinal multivariate analyses of microstructural metrics within individual tract segments. The anatomically heterogeneous start and end regions were excluded[50], yielding 80 segments per tract.

For each of these segments, values from all subjects and time points were extracted. Longitudinal data for each metric were analyzed using linear mixed models (LMMs). The resulting effect matrices were subjected to principal component analysis (PCA), and the multivariate patterns were summarized using PC scores and loadings (RM-ASCA[+],[29]). Correlations between the latent microstructural process (PC1) and DBT learning rates were calculated only for tract segments that showed significant changes in PC1 following motor learning (MRI1 vs. MRI3 and MRI2 vs. MRI3), but not during the non-training control phase (MRI1 vs. MRI2). Lastly, we examined the dimensionality of white matter plasticity, with the hypothesis that correlated changes indicate a common underlying neural network. We also explored whether white matter plasticity relates to previously observed neocortical microstructural plasticity in the same sample[33].

nitrc.org/projects/noddi_toolbox). NODDI models the diffusion signal as a mixture of three compartments: restricted (intra-neurite), hindered (extra-neurite), and isotropic (free water) diffusion. From this model, we extracted three key metrics: the free water fraction (FISO), capturing the proportion of isotropically diffusing water (e.g., cerebrospinal fluid); the intracellular volume fraction (FICVF, commonly referred to as neurite density index, NDI), quantifying the fraction of tissue volume occupied by neurites and other cell processes; and the orientation dispersion index (ODI), reflecting the angular variability of neurite orientation within each voxel[17,22].

**Processing of multiparameter maps.** Quantitative maps based on the MPM protocol were generated in MATLAB (The MathWorks Inc., Natick, MA, USA) using the hMRI toolbox (v0.2.0[78]) within SPM12 (www.fil.ion.ucl.ac.uk/spm), following standard MPM processing steps[35]. Maps were corrected for B1+ and B1− inhomogeneities using dedicated calibration scans acquired prior to each weighted scan: B1+ via

the double-angle method and B1- via receive-sensitivity field maps[72,73]. These calibration maps were used within the hMRI pipeline to correct all quantitative parameter maps for B1+ and B1- bias.

Within the toolbox, the multi-echo data were modeled using the Ernst equation[24], accounting for T1 dependencies, RF spoiling, and B1+ variations[79,80], to generate quantitative maps of magnetization transfer saturation (MTsat), longitudinal relaxation rate (R1 = 1/T1), and proton density (PD). PD maps were further corrected for instrumental bias using the B1- field map. The effective transverse relaxation rate (R2*) was computed using the ESTATICS model[81] combining all multi-echo contrasts to improve robustness and signal-to-noise ratio. Finally, all resulting maps were visually inspected for artifacts.

**Aggregate g-ratio mapping.** The aggregate g-ratio, defined as the ratio of the inner to outer axonal radius[26], is commonly interpreted as reflecting relative axonal myelination. Voxelwise aggregate g-ratio maps

were computed by combining NODDI-derived intra-cellular and iso-tropic volume fractions (FICVF, FISO) to estimate fiber volume fraction (FVF), with MT from the MPM protocol serving as a proxy for the myelin volume fraction (MVF) (ref. [63]).

$$g = \sqrt{1 - \frac{MVF}{FVF}} = \sqrt{1 - \frac{\alpha MT}{(1 - \alpha MT)(1 - A_{FISO})A_{FICFV}}} \quad (1)$$

Because MT does not reflect absolute MVF, we applied a sample-specific calibration factor ($\alpha$). Following Cercignani et al[62]. and Ellerbrock and Mohammadi[63], we extracted mean FICVF, FISO, and MT values from the forceps major (JHU atlas, thresholded at 25%) for each participant. For each individual, $\alpha$ was calculated so that the resulting $g$-ratio in the splenium approximated 0.7, consistent with electron microscopy measurements[64]. The median calibration factor across participants ($\alpha = 0.10$) was then used to compute whole-brain voxelwise aggregate $g$-ratio maps for all participants.

## White matter tract identification and tractometry

Microstructural WM characteristics along selected fiber tracts-of-interest (TOIs) were extracted using TractSeg v2.8[31,32]. Based on neurobiological plausibility[1,5,43,44] and prior research on the DBT[27,28,33,49], 29 tracts of interest were identified: middle cerebellar peduncle, bilateral inferior cerebellar peduncles, superior cerebellar peduncles, frontopontine tracts, corticospinal tracts, anterior thalamic radiations, superior thalamic radiations, thalamo-premotor tracts, striato-fronto-orbital tracts, striatopremotor tracts, superior longitudinal fascicles I-III, corpus callosum (rostrum, genu, rostral body, anterior midbody).

For each participant, preprocessed diffusion data from the pre-learning session (MRI2) were rigidly registered to the FA template in MNI space provided with TractSeg, using FSL FLIRT[82], following the recommended procedure (https://github.com/MIC-DKFZ/TractSeg/). Diffusion data then underwent constrained spherical deconvolution (CSD) using MRtrix3[83,84], followed by neural network-based bundle recognition using CSD peaks to generate Tract Orientation Maps (TOMs)[32]. Finally, probabilistic bundle-specific tractography was performed using TOM peaks and 5000 streamlines.

Tractometry collapses tract-wise information into a relatively small set of averages[59]. A centroid line was first computed for each TOI, then divided into 98 equidistant segments[30] (Fig. 3). Next, each point on all streamlines belonging to a TOI was assigned to a specific segment based on its Euclidean distance to the centroid line[30].

To ensure accurate and anatomically consistent mapping of micro-structural measures to identical locations across time points within each subject, NODDI maps from MRI1 and MRI3 were linearly coregistered to the MRI2 space, and subsequently projected onto the previously segmented TOIs as described above. Similarly, MT maps for each measurement point were aligned to the respective native diffusion space using boundary-based intermodal registration via a T1-weighted anatomical scan[85]; the resulting transformation was then applied to the remaining MPM maps. Finally, MPM microstructural metrics were projected onto the TOIs, as done with the NODDI data.

## Statistical analysis of behaviorally relevant microstructural plasticity

In their seminal review, Thomas and Baker[39] emphasize that a key criterion for interpreting experience-induced plasticity is specificity—demonstrating that changes are unique to the trained group and not observed in untrained individuals. Since our design lacks a traditional control group, we adopted a rigorous multi-step strategy to ensure that the observed latent micro-structural changes are indeed training-related and behaviorally meaningful (Fig. 3). Specifically, we applied Repeated-Measures Analysis of Variance Simultaneous Component Analysis+ (RM-ASCA+[29];) to detect multi-variate changes in 80 segments per tract, excluding the anatomically het-erogeneous start and end regions of each tract[50]. Only tract segments

showing significant latent microstructural changes over time (captured by the first principal component, PC1) *and* correlating with individual learning rates in the DBT were retained for further interpretation. The full workflow is summarized in Fig. 7 and detailed below.

## Linear mixed modelling (LMM) and Repeated-measures analysis of variance simultaneous component analysis (RM-ASCA+)

To assess multivariate changes in white matter microstructure within the TOIs, we employed RM-ASCA+, a recently developed statistical method that combines univariate linear mixed models (LMMs) with multivariate dimension reduction[29]. This approach enables the generation of both score and loading plots for visualizing complex multivariate effects over time.

RM-ASCA+ proceeds in two main steps. First, LMMs were fitted separately for each imaging modality and tract segment to model changes from baseline (MRI1) to subsequent time points (MRI2 and MRI3). In each model, time was specified as a fixed effect, and participant ID was included as a random intercept to account for the dependence between measure-ments taken on the same experimental unit:

$$outcome \sim \text{time} + (1|\text{ID}) \quad (2)$$

Second, the fixed-effect matrices derived from univariate LMMs were subjected to principal component analysis (PCA), producing principal component (PC) scores and loadings[29]. Loadings indicate how strongly each imaging modality contributes to the temporal trajectory of the PC, with positive (or negative) loadings reflecting higher (or lower) values at time points associated with high PC scores.

Robustness of the RM-ASCA+ results was assessed via nonparametric bootstrapping (1000 runs), and 95% confidence intervals (CIs) were com-puted to estimate uncertainty. A statistically significant multivariate change was concluded when CIs for PC scores at MRI3 did *not* overlap with those at MRI1 *and* MRI2[29]. Similarly, significant effects in loadings were inferred when their CIs did not include zero.

Unless otherwise specified, analyses focused on the first PC (PC1), which captured the dominant pattern of plasticity over time within a given tract segment. To minimize the influence of intersubject anatomical variability, statistical analyses were restricted to the middle 80 segments of each tract, excluding the more heterogeneous start and end regions[50].

LMM and RM-ASCA+ analysis were performed in $R$ (v4.2.2) using the lme4 (v1.1-35.1[86]) and ALASCA (v1.0.17[87]) packages.

## Correlated change between white matter microstructure and behavior

In tract segments where PC1 changed after motor learning (MRI1 vs. MRI3 *and* MRI2 vs. MRI3), but *not* during the control phase (MRI1 vs. MRI2), we looked at how these changes related to behavior. To this end, PC1 factor scores were obtained for each subject and measurement point using the ALASCA::predict_scores() function[87]. Because two time intervals are rele-vant for capturing motor learning effects (MRI1–MRI3 and MRI2–MRI3), rank correlation coefficients were computed for both intervals and then combined using the method proposed by Olkin and Pratt[40]. The corre-sponding $p$-values were calculated as described by Schulze[41].

All analyses were conducted in $R$ (v4.2.2) using the ALASCA (v1.0.17[87]) and metacor (v1.0-2.1[88]) packages.

## Monte Carlo analysis

To remind the reader, 29 tracts were analyzed, using 80 segments per tract, yielding a total of 2320 segments. Multiple comparisons are a critical con-cern in such high-dimensional data, but standard $p$-value–based corrections are not applicable in RM-ASCA+[29,87]. To ensure both plasticity and beha-vioral relevance, a segment was considered significant only if it met four stringent criteria: (i) no latent change during the control interval (MRI1–MRI2), (ii) significant learning-related change from MRI1 to MRI3, (iii) significant change from MRI2 to MRI3, and (iv) correlation of latent white matter changes with the behavioral learning rate.

To estimate the probability of observing the number of significant segments purely by chance, we conducted a Monte Carlo simulation under a conservative null model. In 10,000 iterations, four intercorrelated $z$-scores ($\rho = 0.5$) were generated for each of the 2,320 tract segments under the null hypothesis of no true effect ($\mu = 0$, $\sigma = 1$)[89]. For each segment, the least extreme $z$-score (i.e., the smallest in absolute value) was retained, reflecting the requirement that all four criteria be simultaneously satisfied. A segment was classified as a false positive if this retained $z$-score exceeded $|z| > 1.96$. The Monte Carlo $p$-value was defined as the proportion of simulated datasets in which nine or more such false positives occurred—matching the pattern observed in our empirical data—thus providing an empirical estimate of the likelihood of this outcome under the null hypothesis.

### Covariance patterns in white matter plasticity and their correlation with neocortical microstructure

To probe the specificity of our imaging findings, we assessed the dimensionality of latent white matter microstructural changes—captured by PC1 factor scores from the RM-ASCA$^+$ analyses—using principal component analysis (PCA). Previous studies demonstrate that PCA is robust even in small samples when assumptions like sampling adequacy, sphericity, and moderate multicollinearity are met[90,91]. Strong intercorrelations among microstructural features across tract sections would indicate that plasticity processes are coordinated, reflecting changes within a shared structural network[56].

We tested whether training-induced white matter microstructural changes were linked to concurrent changes in neocortical microstructural complexity—especially in motor-related areas—within the same sample[33]. Robust linear regression with MM estimation[92] was used to model neocortical ODI changes as predicted by (i) the shared variance across all significant RM-ASCA$^+$ factor scores and ii) a linear combination of RM-ASCA$^+$ factor scores from all behaviorally relevant tract segments (multiple regression). Model significance was evaluated using a pseudo $F$-ratio derived from a robust deviance test comparing the full model to an intercept-only model[93].

To evaluate out-of-sample predictive performance of the regression models, we performed Leave-One-Out Cross-Validation (LOOCV). For each iteration, the model was trained on all participants except one, and the left-out observation was predicted. This procedure was repeated for all participants. Prediction errors were computed as squared differences between observed and predicted ODI values. Model accuracy was quantified using mean squared error (MSE), root mean squared error (RMSE), and a ratio comparing the model's sum of squared errors (SSE_model) to that of a null model predicting the training mean (SSE_null). The ratio 1 - (SSE_model ÷ SSE_null) indicates the reduction in squared error achieved by the model relative to baseline predictions.

All analyses were conducted in $R$ (v4.2.2) using the packages psych (v2.3.3[94]) and robustbase (v0.99-0[95]).

### Reporting summary

Further information on research design is available in the Nature Portfolio Reporting Summary linked to this article.

## Data availability

Numerical source data underlying all figures and main conclusions are provided in Supplementary Data (Excel format). MRI data supporting the findings have not been deposited in a public repository but are available from the corresponding author (N.L., nico1.lehmann@ovgu.de) upon reasonable request for the purpose of reproducing the results. Requests will be considered promptly and without undue restriction, subject to relevant ethical and institutional approvals.

## Code availability

Custom code and mathematical algorithms used in this study are available from the corresponding author (N.L., nico1.lehmann@ovgu.de) upon reasonable request. Requests will be considered promptly and without undue restriction. Code was developed using Unix/Bash, Python 3.10, MATLAB 2020b, and R 4.2.2, with the relevant packages cited in the manuscript, and can be run in standard Unix-compatible computing environments.

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

## Acknowledgements

This work was supported by the Deutsche Forschungsgemeinschaft (DFG, German Research Foundation) – SFB 1436, Project C01 (project number 425899996).

## Author contributions

N.A.: Investigation, Project administration, Writing - original draft, Writing - Review & Editing, Visualization, J.K.: Methodology, Investigation, Data Curation, Writing - Review & Editing, H.-J.H.: Resources E.D.: Conceptualization, Resources, G.Z.: Conceptualization, Methodology, Writing - Review & Editing, M.T.: Conceptualization, Methodology, Writing - Review & Editing, Supervision, Project administration, Funding acquisition, N.L.: Conceptualization, Methodology, Formal analysis, Investigation, Data Curation, Writing - original draft, Writing - Review & Editing, Visualization.

## Funding

## Competing interests

The authors declare no competing interests.
