## [Transparent Peer Review file · Communications Biology]

Motor learning induces myelin-related white matter changes revealed by MRI-based *In vivo* histology

Corresponding Author: Dr Nico Lehmann

Version 0:

Reviewer comments:

Reviewer #1

(Remarks to the Author)

The present manuscript investigates white matter changes in response to a 4-week balance training paradigm using multiparametric mapping (MPM). I commend the authors for addressing a highly relevant and underexplored area, understanding neuroplasticity in motor learning is important, and the use of MPM and advanced multivariate analyses is innovative. However, there are several major concerns:

1. Claims of biological specificity:

A key issue is the claim that the MPM approach provides biological specificity. While the data are indeed biologically informative and may improve interpretability relative to conventional MRI, biological specificity cannot be claimed for *in vivo* MRI parameters at 3T. Each quantitative MRI metric is influenced by multiple tissue constituents (e.g., T1 depends on water content, macromolecular environment, iron; T2* is affected by iron, myelin, and microstructural geometry). True specificity requires histological validation or multimodal confirmation.

Moreover, there is an ongoing discussion in the field about the distinction between biological sensitivity and biological specificity. Sensitivity means a parameter changes in response to biological variation (e.g., myelination, iron accumulation), but it does not imply it reflects a single underlying property. Specificity would require a one-to-one correspondence between the MRI metric and a biological feature, which cannot be achieved *in vivo*. Rephrasing the claims to highlight sensitivity and relative specificity would strengthen the manuscript and align it with current consensus.

2. RM-ASCA+ and interpretation of latent components:

The use of RM-ASCA+ is, in principle, a reasonable choice for analyzing multivariate repeated measures data. However, interpretation becomes challenging because the SCA components are latent variables, not direct biological measures. Drawing strong biological conclusions from component loadings requires caution. Additionally, given the large number of variables relative to the sample size, there may be concerns about overfitting and robustness of the findings.

3. Specific findings and their interpretation:

The left corticospinal tract (CST) shows a clear, significant effect opposite to the other tracts (Figure 3), yet the manuscript provides no clear rationale for this unexpected finding. Given that the task involves bilateral postural control, such a lateralized effect warrants discussion.

Figure 5 states that "latent white matter microstructural plasticity predicts changes in neocortical neurite orientation." It is unclear why this is framed as a predictive relationship, as it appears to be based on a regression model rather than causal evidence.

4. Choice of tracts and analysis segments:

The selection of tracts of interest (TOIs) is not sufficiently justified. Other tracts relevant for motor learning, such as the superior longitudinal fasciculus (SLF), corpus callosum, and cingulum, should be considered or at least discussed.

The rationale for dividing tracts into 98 equivalent segments is not provided. This arbitrary choice seems to strongly influence the results and should be explained.

5. Sample characteristics:

Only three women were included in the sample. Were any sex-specific analyses conducted, or is the interpretation effectively limited to males? This should be clarified.

6. Suggestions to improve interpretability:

Including correlation plots between behavioral measures and neuroplasticity metrics would substantially aid interpretation and make the reported relationships more transparent.

Summary:

Overall, the manuscript addresses an important research question, and the use of MPM combined with multivariate analysis is a strength. However, claims regarding biological specificity, causal interpretations, and some analytic choices are not sufficiently justified, which undermines the validity of the conclusions.

Reviewer #2

(Remarks to the Author)

This manuscript investigates the biological underpinnings of white matter plasticity during motor learning using a combination of multi-contrast MRI. The authors report tract-segment-specific microstructural changes, primarily driven by myelin-sensitive metrics and g-ratio measurements, that correlated with individual learning rates. They also show that these changes converge into a single latent dimension of white matter plasticity that predicts neocortical reorganization. My comments are listed below:

1. The authors do not explicitly discuss an important trade-off between voxel-wise analysis and tract-segment averaging. If plasticity is confined to a portion of a segment or a subset of streamlines within a tract bundle, the segment-level averaging could dilute sensitivity to those effects.
2. It is unclear whether the authors examined potential spatial overlap between tract segment definitions. Overlap could introduce correlated or redundant information, complicating the interpretation of whether plasticity effects are truly independent across tracts.
3. Because PCA components are directionally arbitrary, decreases in PC1 may not automatically reflect more myelin. Without alignment of component orientation, it is theoretically possible that decreases in PC1 could equally be described as reductions in myelin. A simple way to examine the orientation would be to test whether one of the metrics, e.g., g-ratio measured directly from a tract segment of interest, really increases/decreases across MRI2 and MRI3 as reported.
4. In Figure 5, the latent factor is constructed from all five tracts, including the left CST, which showed an opposite PC1 direction compared with the others. It would be informative to see whether R^2 can be increased if the left CST is removed from this analysis.
5. The cohort consisted of 24 subjects but only 3 were female. It would be interesting to see whether the sensitivity could be increased when the small female subgroup is excluded.
6. Figure 5 is difficult for readers to fully understand without a detailed legend.

Reviewer #3

(Remarks to the Author)

This manuscript presents a sophisticated and timely longitudinal study investigating microstructural white matter (WM) plasticity following motor learning. The authors employ a multi-contrast qMRI approach (MPM & NODDI) combined with advanced tractometry and a novel multivariate statistical framework (RM-ASCA+). The central claim is being able to providing more biologically specific insights into WM plasticity, including evidence for g-ratio changes in humans, which is significant and well-supported by the data. The method is valid and the findings have substantial potential impact for the fields of neuroplasticity, neurorehabilitation, and computational neuroimaging. However, several major points require clarification and additional analysis to fully support the authors' interpretations and to ensure the robustness of the conclusions.

Major concerns:

1. The authors caution that qMRI metrics are sensitive but not perfectly specific. However, some interpretations in the results and discussion could be tempered further. For example, the decrease in FISO in the ATR is interpreted as an "increase in local tissue volume" which is consistent with "glia-related adaptations". While this is a plausible hypothesis, an increase in axonal density or a reduction in extracellular space could produce a similar signal change. The discussion of glia (e.g., astrocytic swelling for PD increases) is interesting but highly speculative. The manuscript would be strengthened by more consistently framing interpretations as plausible hypotheses rather than conclusions, perhaps by adding a paragraph in the discussion explicitly listing the competing biological explanations for the observed signal changes in each key result.
2. The method for calculating the g-ratio is critical to one of the paper's most novel claims. The manuscript states that a sample-specific calibration factor (α) was derived from the forceps major to scale the MT map so that the g-ratio in the splenium is 0.7. This is a common approach but has inherent limitations. How do the key results (e.g., loadings for g-ratio in

FPT, T_PREM, CST) change if the assumed splenial g-ratio is 0.68 or 0.72? This is crucial for establishing the robustness of the findings. Also, why was the forceps major chosen for calibration? Its microstructure (e.g., axon density, packing) differs from the motor tracts where changes are reported. Please justify this choice or discuss it as a limitation.

3. The authors identify that managing multiple comparisons in the RM-ASCA+ framework is complex. The strategy of relying on non-overlapping bootstrapped CIs is reasonable but deserves a more thorough treatment. The analysis involved 35 tracts \times 98 segments = 3430 segments. Only 5 segments showed effects. The probability of finding false positives by chance is a serious concern. The authors must explicitly quantify this risk.

4. The result that the latent WM variable predicts cortical ODI change is one of the key findings. However, given the complexity of the data, a more conservative cross-validated prediction framework (e.g., leave-one-out cross-validation) would provide much stronger evidence for a genuine predictive relationship, rather than just a correlational one in this specific sample.

Minor comments:

1. Please specify the software and version used for the NODDI fitting (presumably the NODDI Matlab Toolbox). For the MPM processing with the hMRI toolbox, please state the key processing steps applied (e.g., were the maps corrected for B1+ and B1- inhomogeneities? This is implied but should be stated explicitly).

2. The paragraph on the left CST (PD/R2* increase) is appropriately cautious. The suggestion that R2* increase could be a consequence of PD increase is insightful; could this be tested directly, for example, by examining the correlation between Δ PD and Δ R2* in that segment across subjects?

3. The citation for the method for g-ratio calculation is Cercignani et al. (2017). However, the described method (using FICVF and FISO to get FVF) aligns more directly with the work of Ellerbrock & Mohammadi (2018, HBM), which should be cited and discussed.

Version 1:

Reviewer comments:

Reviewer #1

(Remarks to the Author)

I thank the the authors for the replies to the concerns that were raised. Multiple concerns resonated also in the comments/suggestions of other authors. Although some are resolved, I feel that major concerns remain.

The first one being related to the wording and statements that are used. In the abstract the authors write about "neocortical reorganization". These are words that do not fit the extend of changes that are observed in such works.

lines 77-79: DTI is not an imaging modality, it is a model that is applied to a MRI protocol, namely diffusion weighted MRI or Diffusion weighed imaging. None of the models or techniques that are proposed in the paper do provide direct biological specificity. DTI is limited in specificity as do the other techniques, but there is no total lack of mapping.

lines 85-93: Overstatement of NODDI biological validity. NODDI parameters are not directly histologically validated in humans, and validation in animals is limited and not always consistent. FICVF does not uniquely represent neurite density. For example, it is also influenced by other factors such as extracellular volume, axon diameter distribution, model assumptions...

Calling it neurite density is an oversimplification..

Figure 1 is misleading in its current state. If the authors refer to "changes" on the left side of the figure, how could they indicate an increase or decrease in the outcome parameters. for example, myelin changes can be of different nature. not only an increase in thickness is possible.

Suggestion to use aggregate g-ratio throughout the manuscript, also in the abstract.

With respect to R1Q3, I appreciate the additional explanation. At the same time, these findings still require some reasoning. Why would the left and right CST "react" modality-specific. What could be a plausible reason for this. In the rebuttal it is not mentioned whether this is now added to the main manuscript.

R1Q4: I thank the reviewers for pointing out that CC and SLF were included in their analyses. What I do not understand is why the negative findings / null findings in these tracts are not discussed as one would anticipate to see changes there based on previous literature. In addition, I understand that a division in 100 segments based on the TractSeg framework is interesting from a practical point of view, but what about the scientific basis for this large number of segments, regardless of total length of the tract? Finally, disregarding the start and end point could mean that we are missing interesting parts. These are the parts that reach close to the GM and therefore are potentially more prone to plasticity.

R1Q5: since also one of the other reviewers asked about this, I suggest to add this information to the text and not limited to a statement in the discussion. I thank the authors for performing these additional analyses.

R1Q6: I thank the authors for adding these correlation plots. These however do not look convincing. Can the authors change the axis ranges to get a better view of the spread. The Y axis rang is narrow, making the trends look steeper? It shows high

scatter relative to trend lines which could indicate substantial individual variability, effect sizes may not be practically meaningful.

Reviewer #2

(Remarks to the Author)

The authors have addressed all my concerns. I have no further questions.

Reviewer #3

(Remarks to the Author)

All my concerns were properly addressed. I have no further questions.

Version 2:

Reviewer comments:

Reviewer #1

(Remarks to the Author)

I thank the authors for the revisions. I have no further major concerns.

Reviewer #1 (Remarks to the Author):

The present manuscript investigates white matter changes in response to a 4-week balance training paradigm using multiparametric mapping (MPM). I commend the authors for addressing a highly relevant and underexplored area, understanding neuroplasticity in motor learning is important, and the use of MPM and advanced multivariate analyses is innovative. However, there are several major concerns:

We thank the reviewer for the thoughtful and constructive comments. We greatly appreciate the recognition of the relevance of our research question and the methodological strengths of our study. The reviewer's suggestions have been important in helping us clarify and strengthen key aspects of the manuscript. Below, we address each point in detail and describe the corresponding revisions made.

R1Q1: Claims of biological specificity: A key issue is the claim that the MPM approach provides biological specificity. While the data are indeed biologically informative and may improve interpretability relative to conventional MRI, biological specificity cannot be claimed for in vivo MRI parameters at 3T. Each quantitative MRI metric is influenced by multiple tissue constituents (e.g., T1 depends on water content, macromolecular environment, iron; T2* is affected by iron, myelin, and microstructural geometry). True specificity requires histological validation or multimodal confirmation.

Moreover, there is an ongoing discussion in the field about the distinction between biological sensitivity and biological specificity. Sensitivity means a parameter changes in response to biological variation (e.g., myelination, iron accumulation), but it does not imply it reflects a single underlying property. Specificity would require a one-to-one correspondence between the MRI metric and a biological feature, which cannot be achieved in vivo. Rephrasing the claims to highlight sensitivity and relative specificity would strengthen the manuscript and align it with current consensus.

R1A1: We fully agree with the reviewer and appreciate this important clarification. In the revised manuscript, all claims of biological specificity have been rephrased to reflect relative specificity, acknowledging that no single in vivo MRI metric at 3T uniquely reflects a single tissue property, in line with current consensus [1–4]. Our interpretations are explicitly based on converging patterns across complementary metrics, and any speculations not fully supported by the data are clearly outlined as such. Although we do not delve into details here, histological validation has been performed for many of the metrics used in this study—for example, NODDI-based metrics [5]. To ensure clarity, we revised the Abstract, Introduction, and Discussion, tempering statements about specificity and highlighting the distinction between biological sensitivity and specificity. Please check.

R1Q2: RM-ASCA+ and interpretation of latent components: The use of RM-ASCA+ is, in principle, a reasonable choice for analyzing multivariate repeated measures data. However, interpretation becomes challenging because the SCA components are latent variables, not direct biological measures. Drawing strong biological conclusions from component loadings requires caution. Additionally, given the large number of variables relative to the sample size, there may be concerns about overfitting and robustness of the findings.

R1A2: We thank the reviewer for this thoughtful comment and fully agree that interpretation of RM-ASCA+ results must acknowledge the latent nature of the extracted components. It is indeed true that these components are not direct biological measures—however, we would argue that this is not a weakness of the method, but one of its key strengths.

As the reviewer rightly noted in the preceding question, individual MRI metrics are influenced by multiple tissue properties and lack perfect specificity. Precisely because of this, there is a growing consensus in the neuroimaging field that multivariate approaches—which explicitly model shared variance across metrics—are more suitable for approximating the underlying neurobiological tissue changes [1,3,6,7]. By integrating these complementary qMRI markers within a longitudinal, multivariate framework, RM-ASCA⁺ captures convergent patterns that better reflect underlying microstructural changes, enabling a more refined *in vivo* characterization of microstructural plasticity during motor learning.

Regarding concerns about sample size and model robustness: PCA has been shown to perform well even in small samples, as long as its assumptions are reasonably met [8]. For example, Casella et al. [9] applied PCA-based plasticity analyses in a study with only $n = 15$ participants. In our case, model fit is strong: PC1 alone consistently explains between 73% and 90% of the total variance. Notably, this exceeds the commonly cited benchmark for component adequacy proposed by Jolliffe [10], who suggests that the cumulative variance explained by all retained components should typically fall within the range of 70–90%. The fact that this threshold is met—and in one case exceeded—by the first principal component alone indicates a remarkably clear underlying structure in the effect matrix and supports the interpretability of the extracted latent dimension.

Moreover, we explicitly used bootstrapping to evaluate the stability and interpretability of the components and loadings. It has been demonstrated that RM-ASCA⁺ yields stable results when bootstrapping is applied [11,12], providing a robust foundation for assessing effect structure and variable importance.

Finally, the component structure is supported by converging univariate results from our linear mixed models (see R1A3 and R2A3). Taken together, we believe the multivariate framework employed here is methodologically appropriate and yields a meaningful representation of training-induced microstructural changes that would not be as readily captured by univariate analyses alone.

R1Q3: Specific findings and their interpretation: The left corticospinal tract (CST) shows a clear, significant effect opposite to the other tracts (Figure 3), yet the manuscript provides no clear rationale for this unexpected finding. Given that the task involves bilateral postural control, such a lateralized effect warrants discussion.

Figure 5 states that "latent white matter microstructural plasticity predicts changes in neocortical neurite orientation." It is unclear why this is framed as a predictive relationship, as it appears to be based on a regression model rather than causal evidence.

R1A3: We appreciate the reviewer's attention to this point. The apparent lateralization of PC1 trajectories reflects tract-specific contributions of different imaging modalities rather than an unexpected lateralized effect.

The loadings indicate how strongly each imaging metric contributes to the temporal trajectory of the first principal component (PC1): positive (or negative) loadings show that the metric changes in the same (or opposite) direction as PC1 over time (see Methods and Figure 3 caption).

- In the right CST, PC1 decreases after training, with a positive loading for *g*-ratio. Since PC1 and *g*-ratio move together, *g*-ratio also decreases, potentially indicating increased relative axonal myelination — an expected training-related effect [13].

- In the left CST, PC1 increases, with the strongest positive loading from R2*, implying that both PC1 and R2* increase together over time.

These findings are corroborated by univariate linear mixed models (LMMs), confirming that the effects are modality-specific:

- R2* in CST_left: $b = 0.46$, $p = .03$
- g-ratio in CST_right: $b = -0.41$, $p = .04$

Regarding Figure 6 (formerly Figure 5), we clarified that “predicts” refers to a statistical association, not causality. The revised caption reads:

“Figure 6. Negative relationship between latent white matter plasticity (PC1 score from Table 1) and post-training changes in neocortical neurite orientation dispersion index (ODI; Ref. [31]), estimated using robust simple regression [44]. The gray band represents the 95% confidence interval of the regression line. Below, tract reconstructions indicate the five white matter segments contributing to the latent variable. The cortical surface (left) highlights regions showing significant post-training ODI increases (red; $p < .05$, corrected [31]).”

In summary, the divergent PC1 trajectories across left and right CST are biologically plausible and reflect tract-specific contributions of different MRI metrics rather than lateralization artifacts.

R1Q4: Choice of tracts and analysis segments: The selection of tracts of interest (TOIs) is not sufficiently justified. Other tracts relevant for motor learning, such as the superior longitudinal fasciculus (SLF), corpus callosum, and cingulum, should be considered or at least discussed.

The rationale for dividing tracts into 98 equivalent segments is not provided. This arbitrary choice seems to strongly influence the results and should be explained.

R1A4: We thank the reviewer for this thoughtful comment and the opportunity to clarify our rationale.

Tract selection: Our selection of 29 tracts was hypothesis-driven, focusing on white matter pathways connecting sensorimotor hub regions that are known to be dynamically modulated by balance and motor-skill training (see Methods for references). The SLF and corpus callosum were indeed included in our analyses, though they did not show behaviorally relevant plasticity in our sample. Regarding the cingulum, while it could be involved in complex motor learning, we did not include it in order to maintain a focused hypothesis-driven approach on tracts most directly connected to the sensorimotor network.

Tract segmentation: The division into 98 equivalent segments follows the bundle analytics [14] framework implemented in TractSeg [15,16]. Each tract is initially divided into 100 equidistant points along its trajectory [14]. To reduce partial-volume effects, the first and last points are discarded, resulting in 98 segments. As noted by the TractSeg developer: *“The first and last segment are discarded since they are not as robust. So only 98 are used.”* <https://github.com/MIC-DKFZ/TractSeg/issues/204>

Note that, for statistical analyses, only the middle 80 segments were used to avoid anatomically heterogeneous start and end regions, consistent with previous studies [17].

R1Q5: Sample characteristics: Only three women were included in the sample. Were any sex-specific analyses conducted, or is the interpretation effectively limited to males? This should be clarified.

R1A5: We thank the reviewer for raising this important point. Indeed, our sample was predominantly male (21 males, 3 females), reflecting recruitment constraints rather than an a priori exclusion criterion.

To address potential sex effects, we performed reanalyses excluding the three female participants. The results were largely consistent with the full sample, with minor differences: some measures showed slightly greater stability during the control period and/or somewhat more pronounced neuroplastic effects following training. These observations indicate that inclusion of the small female subgroup did not materially alter our conclusions.

We have now explicitly noted this in the Discussion to clarify the cohort composition and its implications for interpretation: *“Other limitations include the small number of female participants (n = 3), so results predominantly reflect males...”*

R1Q6: Suggestions to improve interpretability: Including correlation plots between behavioral measures and neuroplasticity metrics would substantially aid interpretation and make the reported relationships more transparent.

R1A6: We thank the reviewer for this helpful suggestion. To enhance interpretability and transparency, Figure 4 has been revised to include rank correlation scatterplots illustrating the associations between behavioral measures and neuroplasticity metrics. These plots correspond to the correlations that were subsequently aggregated across time intervals using the Olkin–Pratt method [18]. We believe that this addition makes the reported relationships more intuitive and accessible to readers.

Summary: Overall, the manuscript addresses an important research question, and the use of MPM combined with multivariate analysis is a strength. However, claims regarding biological specificity, causal interpretations, and some analytic choices are not sufficiently justified, which undermines the validity of the conclusions.

We thank the reviewer for the thoughtful summary and for the constructive feedback throughout the review. We fully agree that careful interpretation is warranted regarding biological specificity, causal inferences, and analytic choices. In the revised manuscript, we have addressed these points by tempering claims, clarifying that reported relationships reflect statistical associations rather than causality, and providing detailed justifications for tract selection, segmentation, and the RM-ASCA⁺ analysis framework. Together with additional robustness checks, new visualizations, and clarifications, we believe the manuscript is now substantially improved in clarity, interpretability, and scientific rigor.

Reviewer #2 (Remarks to the Author):

This manuscript investigates the biological underpinnings of white matter plasticity during motor learning using a combination of multi-contrast MRI. The authors report tract-segment-specific microstructural changes, primarily driven by myelin-sensitive metrics and g-ratio measurements, that correlated with individual learning rates. They also show that these changes converge into a single latent dimension of white matter plasticity that predicts neocortical reorganization. My comments are listed below:

We thank the reviewer for the accurate summary of our study. We appreciate the recognition of our main findings and the constructive efforts made to improve the clarity, interpretation, and rigor of the manuscript.

R2Q1: The authors do not explicitly discuss an important trade-off between voxel-wise analysis and tract-segment averaging. If plasticity is confined to a portion of a segment or a subset of streamlines within a tract bundle, the segment-level averaging could dilute sensitivity to those effects.

R2A1: Thank you for highlighting this important trade-off. We agree that segment-level averaging in tractometry can dilute highly localized plasticity effects, especially if they are confined to very small subregions. However, this reflects a broader trade-off between spatial specificity and statistical power in MRI analyses [19].

Voxel-wise methods (e.g., VBA, TBSS) offer high spatial resolution but are vulnerable to misregistration errors, low signal-to-noise ratio, and severe multiple comparison penalties, which limit statistical power [20]. ROI averaging improves sensitivity by reducing noise and the number of comparisons but risks masking focal changes. Tractometry, particularly along-tract profiling, provides a practical compromise: it maintains meaningful anatomical specificity while enhancing statistical power and robustness [15,16,21,22]. This approach is increasingly favored over pure voxel-wise analyses for white matter, given the challenges noted above.

In the revised Discussion, we now explicitly address this trade-off:

“Segment-level averaging in tractometry, while potentially diluting highly localized effects confined to small subregions, represents a practical and balanced compromise: it enhances statistical power and reduces noise compared with voxel-wise analyses, while retaining meaningful anatomical specificity [28–30,52,60]. Pure voxel-wise methods offer higher spatial resolution but face challenges such as misregistration, low signal-to-noise ratio, and severe multiple comparison penalties [32,62].”

R2Q2: It is unclear whether the authors examined potential spatial overlap between tract segment definitions. Overlap could introduce correlated or redundant information, complicating the interpretation of whether plasticity effects are truly independent across tracts.

R2A2: We thank the reviewer for this insightful and important comment. Indeed, the potential for spatial overlap between tract definitions—particularly in regions like the internal capsule, corona radiata, and cerebral peduncle—raises a valid concern. Such overlap could lead to correlated microstructural changes across tracts that do not necessarily reflect distinct plasticity processes.

To evaluate this possibility, we generated 3D binary masks for all tracts and their segments (left panel of the Figure below exemplifies this for the corticospinal tracts) showing significant effects and computed pairwise intersection masks (i.e., $FPT_R \cap T_PREM_R$, $FPT_R \cap CST_R$, $FPT_R \cap ATR_R$, $T_PREM_R \cap CST_R$, $T_PREM_R \cap ATR_R$, $CST_R \cap ATR_R$). The right-hand side of the Figure below shows the partial overlap between the right T_PREM (segments 73–74) and the right FPT (segment 50) within the internal capsule, viewed from a posterior perspective. All other tract pairs showed no spatial overlap.

These findings suggest that the observed effects are largely tract-specific and not primarily driven by shared anatomical regions. We have added a note to the Discussion acknowledging the partial overlap between right T_PREM and right FPT to appropriately qualify this aspect of our interpretation (see below). Notably, the internal capsule—where the overlap was found—is a well-known bottleneck region in the brain, where multiple fiber bundles converge and run in parallel orientation [23]. Such anatomical convergence presents a persistent challenge for tractography, and even state-of-the-art methods cannot fully resolve these ambiguities. Therefore, some degree of spatial overlap in this region is difficult to avoid and should be considered when interpreting the tract-specificity of the observed effects.

The following statement has been added to the Discussion section:

"Other limitations include [...] potential overlap of significant FPT and T_PREM segments in the internal capsule—a classical "bottleneck" region [61]—limiting tract-specific interpretation."

R2Q3: Because PCA components are directionally arbitrary, decreases in PC1 may not automatically reflect more myelin. Without alignment of component orientation, it is theoretically possible that decreases in PC1 could equally be described as reductions in myelin. A simple way to examine the orientation would be to test whether one of the metrics, e.g., g-ratio measured directly from a tract segment of interest, really increases/decreases across MRI2 and MRI3 as reported.

R2A3: Thank you for raising this important point about the arbitrary directionality of PCA components. As noted in one of our earlier responses (R1A3), the biological interpretation of PC1 depends on the loadings of the individual imaging metrics. Positive loadings indicate that a metric changes in the same direction as PC1 over time; negative loadings indicate the opposite (see Methods and Figure 3 caption). This effectively anchors the component's sign to biologically meaningful change.

To verify this alignment, we present below univariate linear mixed model results (change from MRI1 to MRI3) for the metrics with the strongest loadings on each multivariate effect. For example, when *g*-ratio shows a positive loading and PC1 decreases, the *g*-ratio should also decrease—consistent with potentially increased relative myelination. This pattern is reflected in the *negative* regression coefficients for *g*-ratio in CST_R and FPT_R.

tract segment	Metric	b	p
ATR (RH)	FISO	-0.39	.05
T_PREM (RH)	FISO	-0.48	.02
FPT (RH)	g -ratio	-0.29	.06
CST (RH)	g -ratio	-0.41	.04
CST (LH)	R2*	0.48	.02

Thus, although PCA component direction is mathematically arbitrary, the combination of loadings and supporting univariate results ensures that decreases or increases in PC1 correspond meaningfully to the underlying metrics. We appreciate the reviewer’s attention to this nuance and hope this clarifies our approach.

R2Q4: In Figure 5, the latent factor is constructed from all five tracts, including the left CST, which showed an opposite PC1 direction compared with the others. It would be informative to see whether R² can be increased if the left CST is removed from this analysis.

R2A4: We thank the reviewer for this thoughtful suggestion. Removing the left CST from the PCA would alter the latent factor scores and could affect the R² when predicting neocortical ODI changes. However, as shown in Table 1 of the manuscript, the left CST has the lowest absolute loading on the latent factor, indicating limited influence. Note that its negative loading direction also aligns with the temporal trajectory of PC1 (i.e., decreasing for the left CST while increasing for the other tracts; see Figure 3 of the manuscript).

Moreover, the full regression model with all five tracts explains 36% of the variance in ODI change, compared to 19% explained by the latent variable alone. This suggests that the left CST contributes criterion-relevant variance not captured by the latent variable.

Thus, while excluding the left CST might slightly change R², it would likely reduce the latent variable’s representativeness in terms of neocortical ODI changes. We appreciate the reviewer’s insight and can provide this supplementary analysis upon request.

R2Q5: The cohort consisted of 24 subjects but only 3 were female. It would be interesting to see whether the sensitivity could be increased when the small female subgroup is excluded.

R2A5: We thank the reviewer for this valuable suggestion. We conducted the requested reanalyses excluding the three female participants. The results remained largely consistent, with some measures showing slightly increased stability during the control period and/or more pronounced neuroplastic effects following training.

These findings indicate that inclusion of the small female subgroup does not materially affect the robustness of our conclusions. We have added this consideration to the Discussion to acknowledge the cohort composition and its implications:

“Other limitations include the small number of female participants (n = 3), so results predominantly reflect males, which should be considered when generalizing the findings, ...”.

R2Q6: Figure 5 is difficult for readers to fully understand without a detailed legend.

R2A6: We thank the reviewer for this helpful observation. We assume the reviewer refers to the figure caption; accordingly, we have added a detailed caption to clarify the components and interpretation of Figure 5 (which is now Figure 6).

“Figure 6. Negative relationship between latent white matter plasticity (PC1 score from Table 1) and post-training changes in neocortical neurite orientation dispersion index (ODI; Ref. [31]), estimated using robust simple regression [44]. The gray band represents the 95% confidence interval of the regression line. Below, tract reconstructions indicate the five white matter segments contributing to the latent variable. The cortical surface (left) highlights regions showing significant post-training ODI increases (red; p < .05, corrected [31]).”

Reviewer #3 (Remarks to the Author):

This manuscript presents a sophisticated and timely longitudinal study investigating microstructural white matter (WM) plasticity following motor learning. The authors employ a multi-contrast qMRI approach (MPM & NODDI) combined with advanced tractometry and a novel multivariate statistical framework (RM-ASCA+). The central claim is being able to providing more biologically specific insights into WM plasticity, including evidence for g-ratio changes in humans, which is significant and well-supported by the data. The method is valid and the findings have substantial potential impact for the fields of neuroplasticity, neurorehabilitation, and computational neuroimaging. However, several major points require clarification and additional analysis to fully support the authors' interpretations and to ensure the robustness of the conclusions.

We sincerely thank the reviewer for the thoughtful and encouraging remarks. We greatly appreciate the recognition of the novelty, methodological rigor, and potential impact of our study. We are also grateful for the constructive points raised, which have helped us to clarify and strengthen the manuscript.

Major concerns:

R3Q1: The authors caution that qMRI metrics are sensitive but not perfectly specific. However, some interpretations in the results and discussion could be tempered further. For example, the decrease in FISO in the ATR is interpreted as an “increase in local tissue volume” which is consistent with “glia-related adaptations”. While this is a plausible hypothesis, an increase in axonal density or a reduction in extracellular space could produce a similar signal change. The discussion of glia (e.g., astrocytic swelling for PD increases) is interesting but highly speculative. The manuscript would be strengthened by more consistently framing interpretations as plausible hypotheses rather than conclusions, perhaps by adding a paragraph in the discussion explicitly listing the competing biological explanations for the observed signal changes in each key result.

R3A1: We thank the reviewer for this insightful and constructive comment. We fully agree that our interpretations should remain hypothesis-oriented rather than conclusive. In response to this and Reviewer #1's related feedback, we have carefully revised the wording throughout the manuscript—particularly in the Abstract, Introduction, and Discussion—to temper claims regarding the biological specificity of qMRI metrics. Please check.

While we initially considered adding a separate paragraph listing alternative biological interpretations, we found that this approach disrupted the flow of the Discussion. Instead, we have embedded these nuances directly into the relevant result interpretations, ensuring that speculative explanations are now clearly presented as plausible hypotheses rather than definitive conclusions. Wherever interpretive extensions are made, they are explicitly identified as such and supported by references to relevant empirical or theoretical work—ensuring that our speculations are grounded in established literature rather than conjecture. We hope these revisions strike an appropriate balance between interpretive clarity, scientific caution, and readability.

In the case of the ATR finding, we note that the observed decrease in FISO was accompanied by a trend toward a decrease in NODDI's intracellular volume fraction (FICVF) (Figure 3). Because FICVF is generally thought to reflect a proxy for neurite density [5,24], an *increase* in axonal density as suggested by the reviewer is unlikely to explain this pattern. Instead, glia-related processes or a reduction in extracellular space appear to be more plausible contributors. The section in question now reads:

“The strength of the multivariate approach is particularly evident in interpreting behaviorally relevant plasticity in the right anterior thalamic radiation (ATR_R). Here, a decrease in the free water fraction might

indicate enhanced tissue integrity and/or reduced extracellular space. This pattern could in theory also reflect glia-related adaptations [17,52], since the tendency towards decreased intracellular volume fraction might reflect an expansion of neuroglia [17,18]; however, this interpretation remains tentative given wide confidence intervals.”

Regarding the R2*/PD effects in the left CST, we believe that the discussion of biological explanations is relatively cautious, as the reviewer contends in R3Q6. Nonetheless, we now more explicitly acknowledge that parts of these interpretations remain speculative.

“While these changes correlated with motor learning, their biological interpretation remains uncertain. An increase in PD could plausibly reflect transient cellular phenomena such as astrocytic swelling [54,55]. The concurrent R2 increase might relate to iron-associated processes—either glial changes involving iron deposits in neuroglial somata [22] or vascular alterations linked to heme-bound iron [56]. Alternatively, the rise in R2* might result from increased PD itself, since elevated water content can enhance magnetic susceptibility and accelerate relaxation.”*

R3Q2: The method for calculating the g -ratio is critical to one of the paper's most novel claims. The manuscript states that a sample-specific calibration factor (α) was derived from the forceps major to scale the MT map so that the g -ratio in the splenium is 0.7. This is a common approach but has inherent limitations. How do the key results (e.g., loadings for g -ratio in FPT, T_PREM, CST) change if the assumed splenial g -ratio is 0.68 or 0.72? This is crucial for establishing the robustness of the findings. Also, why was the forceps major chosen for calibration? Its microstructure (e.g., axon density, packing) differs from the motor tracts where changes are reported. Please justify this choice or discuss it as a limitation.

R3A2: We thank the reviewer for this insightful comment regarding the calibration method used in our g -ratio estimation. We fully agree that evaluating the robustness of our findings with respect to the assumed splenial g -ratio value is important [25].

As suggested, we recalculated the g -ratio maps using alternative splenial reference values of 0.68 and 0.72, resulting in calibration factors of $\alpha = 0.11$ and $\alpha = 0.0955$, respectively (compared to $\alpha = 0.10$ for 0.70). We then recomputed the g -ratio maps and re-examined the relevant metrics. Cross-sectional correlation analyses in the tract segments showing significant g -ratio loadings (CST_right, FPT_right, T_PREM_right) revealed near-perfect correspondence between the original and recalibrated maps (see Figure below). While Ellerbrock and Mohammadi [25] reported somewhat lower correspondence when testing a broader g -ratio range (0.6–0.8), our results indicate that the present conclusions are robust to plausible variations in the calibration constant.

Regarding the choice of the forceps major (splenium) as the calibration region, we appreciate the reviewer’s concern that its microstructure differs from motor tracts. Our rationale is twofold:

- 1) Empirical precedent: The true in vivo g -ratio of healthy white matter is not precisely known [26]. However, as noted by Mohammadi et al. [27] and Ellerbrock and Mohammadi [25], microscopic g -ratio values around 0.7 have been observed in large-diameter callosal axons from human post mortem tissue [28]. This provides a biologically grounded rationale for using the splenium as a calibration region.
- 2) Comparability across studies: Using a standard calibration point—typically a splenial g -ratio of 0.70—facilitates methodological consistency and cross-study comparability [29], which also motivated our choice.

Taken together, the calibration choice and range tested here fall well within accepted practice and do not materially affect our main findings.

The following sentences have now been added to the revised Discussion section:

“Another methodological consideration concerns the calibration of the g -ratio map, which was anchored to a splenial reference value of 0.70 [63,64]. While this approach is consistent with prior studies and supported by histological evidence [65], the true in vivo g -ratio may vary across white matter regions [66]. Sensitivity analyses using recalibrated maps with reference values between 0.68 and 0.72 yielded near-perfect correlations with the original maps, suggesting that our conclusions are robust within this physiologically plausible range.”

R3Q3: The authors identify that managing multiple comparisons in the RM-ASCA+ framework is complex. The strategy of relying on non-overlapping bootstrapped CIs is reasonable but deserves a more thorough treatment. The analysis involved 35 tracts \times 98 segments = 3430 segments. Only 5 segments showed effects. The probability of finding false positives by chance is a serious concern. The authors must explicitly quantify this risk.

R3A3: We thank the reviewer for raising this extremely important point regarding the risk of false positives due to multiple comparisons in the RM-ASCA+ framework.

Before addressing the substantive concern, we would like to clarify the exact number of tract segments analyzed. In the original manuscript, we inadvertently stated that 35 tracts were included. Three of these (T_PREF, T_PREC, and ST_PREC in both hemispheres) could not be reliably segmented using TractSeg and were therefore excluded, resulting in a total of 29 tracts analyzed. Furthermore, only the central 80 segments per tract were included, consistent with previous tractometry studies [17], to avoid the anatomically heterogeneous start and end regions. Thus, the total number of analyzed segments was 2,320 (29 tracts × 80 segments). These corrections have been implemented in the revised manuscript.

Turning to the main issue, we fully agree that multiple comparisons represent a key statistical challenge in this type of high-dimensional analysis. As correctly noted, our inference within RM-ASCA⁺ was based on non-overlapping 95% BCa bootstrap confidence intervals, following the original method [11,12]. Although this approach does not yield *p*-values that can be directly corrected for multiple testing, we employed several stringent safeguards to minimize the likelihood of false positives. Specifically, a tract segment was only considered significant if all four of the following independent criteria were simultaneously satisfied:

1. No latent microstructural change during the control interval (MRI1–MRI2);
2. A significant learning-related change (non-overlapping 95% BCa CIs) from MRI1–MRI3;
3. A significant learning-related change from MRI2–MRI3; and
4. A significant correlation between the latent microstructural change during learning and the behavioral learning rate (stabilometer performance).

This multi-criterion approach imposes a high threshold for declaring significance, thereby markedly reducing the probability of false positives.

To explicitly quantify this risk, we conducted a Monte Carlo simulation under a conservative null model. In 10,000 iterations, four intercorrelated *z*-scores ($\rho = 0.5$) were generated for each of the 2,320 segments under the null hypothesis of no true effect ($\mu = 0, \sigma = 1$) [30]. For each segment, the least extreme *z*-score (i.e., the smallest in absolute value) was retained, reflecting the requirement that all four criteria be simultaneously met. A segment was classified as a false positive if this retained *z*-score exceeded $|z| > 1.96$. The Monte Carlo *p*-value was defined as the proportion of simulated datasets in which nine or more such false positives occurred—matching the pattern observed in our empirical data (ATR_R: 3 neighbouring segments, T_PREM: 2 neighbouring segments, CST_L: 2 neighbouring segments, CST_R: 1 segment, FPT_R: 1 segment).

This simulation yielded a p -value of 0.01, indicating that the likelihood of observing the empirical pattern purely by chance is very low, even under a correlated null model. Importantly, even when the potentially overlapping right FPT segment was excluded (see R2A2), the result remained statistically significant ($p = 0.028$). These analyses have now been added to the revised manuscript to substantiate the robustness of our findings.

R3Q4: The result that the latent WM variable predicts cortical ODI change is one of the key findings. However, given the complexity of the data, a more conservative cross-validated prediction framework (e.g., leave-one-out cross-validation) would provide much stronger evidence for a genuine predictive relationship, rather than just a correlational one in this specific sample.

R3A4: We thank the reviewer for this important suggestion. As recommended, we performed Leave-One-Out Cross-Validation (LOOCV) using robust linear regression models (lmrob, robustbase R package) to evaluate out-of-sample predictive performance. For each iteration, the model was trained on all participants except one, and the left-out observation was then predicted. This process was repeated for all subjects. Prediction errors were computed as squared differences between observed and predicted ODI values. Model accuracy was quantified via mean squared error (MSE), root mean squared error (RMSE), and a ratio comparing the model’s sum of squared errors (SSE_model) to that of a null model predicting the training mean (SSE_null). This ratio indicates the reduction in squared error achieved by the model relative to baseline predictions.

Two models were tested:

- Model 1 (latent variable) – using the single latent white matter plasticity score as predictor;
- Model 2 (five tract-specific predictors) – using the five individual tract components as predictors.

model	no. of predictors	R^2 (original model)	LOOCV MSE	LOOCV RMSE	$1 - (SSE_{model} \div SSE_{null})$
Version 1	1	.19	22.36	4.73	0.07
Version 2	5	.36	26.46	5.14	-0.10

As shown in the table, the latent variable model slightly outperformed the null model (LOOCV ratio = 0.07), suggesting modest out-of-sample predictive power. In contrast, the five-predictor model exhibited a negative ratio, consistent with overfitting.

Taken together, these results support the reviewer’s view that the observed association between the latent white matter component and cortical ODI change reflects a sample-specific correlation rather than a strong, generalizable predictive relationship. We have included these analyses in the revised manuscript and accordingly tempered our interpretation of the ODI–white matter findings.

Results: *“Leave-One-Out Cross-Validation (LOOCV) results suggest that while both robust regression models capture meaningful relationships in this sample, their predictive generalizability to independent data may be limited (Table 2).”*

Discussion: *“Notably, these distributed white matter changes also correlated with neocortical plasticity in the same sample [31], further suggesting coordinated, network-level adaptation, although LOOCV analyses indicate these associations may be sample-specific.”*

Minor comments:

R3Q5: Please specify the software and version used for the NODDI fitting (presumably the NODDI Matlab Toolbox). For the MPM processing with the hMRI toolbox, please state the key processing steps applied (e.g., were the maps corrected for B1+ and B1- inhomogeneities? This is implied but should be stated explicitly).

R3A5: We thank the reviewer for this helpful comment and the opportunity to clarify the software versions and processing steps used in both the NODDI and MPM pipelines.

Regarding NODDI:

As stated in the revised Methods section: *“Microstructural parameter maps were estimated using the NODDI model [17], applied to the preprocessed multishell diffusion data ($b = 0, 1,000, 2,000, \text{ and } 3,000 \text{ s/mm}^2$) using the NODDI Matlab Toolbox v1.0.1 (http://nitrc.org/projects/noddi_toolbox).”*

Regarding B1⁺ and B1⁻ corrections:

Transmit and receive field calibration scans were acquired prior to each weighted scan:

“Transmit and receive field calibration scans were performed prior to each weighted scan: B1- mapping: 56 sagittal slices, $TR = 4.1 \text{ ms}$, $TE = 1.98 \text{ ms}$, $FOV = 230 \times 230 \text{ mm}$; B1+ mapping: 24 sagittal slices (5 mm thickness), $TR = 2,000 \text{ ms}$, $TE1 = TE2 = 14 \text{ ms}$, flip angles = $90^\circ, 120^\circ, 60^\circ, 135^\circ, 45^\circ$ [73,74].”

The revised “Processing of multiparameter maps” section now explicitly states the key processing steps and the application of B1+ and B1- corrections:

“Quantitative maps based on the MPM protocol were generated in MATLAB (The MathWorks Inc., Natick, MA, USA) using the hMRI toolbox (v0.2.0 [79]) within SPM12 (www.fil.ion.ucl.ac.uk/spm), following standard MPM processing steps [33]. Maps were corrected for B1+ and B1- inhomogeneities using dedicated calibration scans acquired prior to each weighted scan: B1+ via the double-angle method and B1- via receive-sensitivity field maps [73,74]. These calibration maps were used within the hMRI pipeline to correct all quantitative parameter maps for B1+ and B1- bias.

Within the toolbox, the multi-echo data were modeled using the Ernst equation [20], accounting for T1 dependencies, RF spoiling, and B1+ variations [80,81], to generate quantitative maps of magnetization transfer saturation (MTsat), longitudinal relaxation rate ($R1 = 1/T1$), and proton density (PD). PD maps were further corrected for instrumental bias using the B1- field map. The effective transverse relaxation rate ($R2^$) was computed using the ESTATICS model [82] combining all multi-echo contrasts to improve robustness and signal-to-noise ratio. Finally, all resulting maps were visually inspected for artifacts.”*

R3Q6: The paragraph on the left CST (PD/ $R2^*$ increase) is appropriately cautious. The suggestion that $R2^*$ increase could be a consequence of PD increase is insightful; could this be tested directly, for example, by examining the correlation between ΔPD and $\Delta R2^*$ in that segment across subjects?

R3A6: We thank the reviewer for this constructive suggestion. To directly test whether the observed $R2^*$ increase in the left CST segment could be explained by concomitant changes in PD, we correlations between ΔPD and $\Delta R2^*$ in the segment showing concurrent parameter changes. The resulting correlation coefficients were modest and did not reach statistical significance ($\Delta MRI2-MRI3$: $r = 0.09$; $\Delta MRI1-MRI3$: $r = 0.33$, $p > 0.1$).

These results indicate that the $R2^*$ increase cannot be attributed to PD-related effects alone. While local changes in tissue water content may have contributed partially to the $R2^*$ signal, the weak-to-moderate association suggests that additional mechanisms—such as subtle iron redistribution, glial activity, or vascular alterations—might have also played a role.

The following changes have been made to the Discussion section:

“Alternatively, the rise in $R2^$ might result from increased PD itself, since elevated water content can enhance magnetic susceptibility and accelerate relaxation. However, given the at best moderate association between PD and $R2^*$ changes in this tract segment ($r \leq 0.33$, n.s.), any such interdependence is likely limited.”*

R3Q7: The citation for the method for g-ratio calculation is Cercignani et al. (2017). However, the described method (using FICVF and FISO to get FVF) aligns more directly with the work of Ellerbrock & Mohammadi (2018, HBM), which should be cited and discussed.

R3A7: You are absolutely correct — the calculation corresponds to variant g3 (Equation 7) from Ellerbrock & Mohammadi (2018, HBM). We thank you for pointing this out and have added the appropriate citation in the revised manuscript.

“Because MT does not reflect absolute MVF, we applied a sample-specific calibration factor (α). Following Cercignani et al. [63] and Ellerbrock and Mohammadi [64], we extracted mean FICVF, FISO, and MT values from the forceps major (JHU atlas, thresholded at 25%) for each participant. For each individual, α was calculated so that the resulting g-ratio in the splenium approximated 0.7, consistent with electron microscopy measurements [65]. The median calibration factor across participants ($\alpha = 0.10$) was then used to compute whole-brain voxelwise g-ratio maps for all participants.”

References

1. Cercignani, M. & Bouyagoub, S. Brain microstructure by multi-modal MRI: Is the whole greater than the sum of its parts? *NeuroImage* **182**, 117–127 (2018).
2. Weiskopf, N., Edwards, L. J., Helms, G., Mohammadi, S. & Kirilina, E. Quantitative magnetic resonance imaging of brain anatomy and in vivo histology. *Nat. Rev. Phys.* **3**, 570–588 (2021).
3. Tardif, C. L. *et al.* Advanced MRI techniques to improve our understanding of experience-induced neuroplasticity. *NeuroImage* **131**, 55–72 (2016).
4. Weiskopf, N., Mohammadi, S., Lutti, A. & Callaghan, M. F. Advances in MRI-based computational neuroanatomy: from morphometry to in-vivo histology. *Current opinion in neurology* **28**, 313–322 (2015).
5. Kraguljac, N. V., Guerreri, M., Strickland, M. J. & Zhang, H. Neurite Orientation Dispersion and Density Imaging in Psychiatric Disorders: A Systematic Literature Review and a Technical Note. *Biol. Psychiatry Glob. Open Sci.* **3**, 10–21 (2023).
6. Lazari, A. & Lipp, I. Can MRI measure myelin? Systematic review, qualitative assessment, and meta-analysis of studies validating microstructural imaging with myelin histology. *NeuroImage* **230**, 117744 (2021).
7. Kamiya, K., Hori, M. & Aoki, S. NODDI in clinical research. *Journal of neuroscience methods* **346**, 108908 (2020).
8. de Winter, J. C. F., Dodou, D. & Wieringa, P. A. Exploratory Factor Analysis With Small Sample Sizes. *Multivariate Behav. Res.* **44**, 147–181 (2009).
9. Casella, C. *et al.* Drumming Motor Sequence Training Induces Apparent Myelin Remodelling in Huntington's Disease: A Longitudinal Diffusion MRI and Quantitative Magnetization Transfer Study. *Journal of Huntington's disease* **9**, 303–320 (2020).
10. Jolliffe, I. T. *Principal Component Analysis* (Springer-Verlag New York Inc, 2002).
11. Jarmund, A. H., Madssen, T. S. & Giskeødegård, G. F. ALASCA: An R package for longitudinal and cross-sectional analysis of multivariate data by ASCA-based methods. *Front. Mol. Biosci.* **9**, 962431 (2022).
12. Madssen, T. S., Giskeødegård, G. F., Smilde, A. K. & Westerhuis, J. A. Repeated measures ASCA+ for analysis of longitudinal intervention studies with multivariate outcome data. *PLoS Comput. Biol.* **17**, e1009585 (2021).
13. Sampaio-Baptista, C. & Johansen-Berg, H. White Matter Plasticity in the Adult Brain. *Neuron* **96**, 1239–1251 (2017).
14. Chandio, B. Q. *et al.* Bundle analytics, a computational framework for investigating the shapes and profiles of brain pathways across populations. *Sci. Rep.* **10**, 17149 (2020).
15. Wasserthal, J., Neher, P. & Maier-Hein, K. H. TractSeg - Fast and accurate white matter tract segmentation. *NeuroImage* **183**, 239–253 (2018).
16. Wasserthal, J., Neher, P. F., Hirjak, D. & Maier-Hein, K. H. Combined tract segmentation and orientation mapping for bundle-specific tractography. *Med. Image Anal.* **58**, 101559 (2019).
17. Huber, E., Mezer, A. & Yeatman, J. D. Neurobiological underpinnings of rapid white matter plasticity during intensive reading instruction. *NeuroImage* **243**, 118453 (2021).
18. Olkin, I. & Pratt, J. W. Unbiased Estimation of Certain Correlation Coefficients. *Ann. Math. Stat.* **29**, 201–211 (1958).
19. Lehmann, N. *et al.* Longitudinal Reproducibility of Neurite Orientation Dispersion and Density Imaging (NODDI) Derived Metrics in the White Matter. *Neuroscience* **457**, 165–185 (2021).
20. van Hecke, W., Leemans, A. & Emsell, L. in *Diffusion Tensor Imaging*, edited by W. van Hecke, L. Emsell & S. Sunaert (Springer New York, 2016), pp. 183–203.
21. Yeatman, J. D., Dougherty, R. F., Myall, N. J., Wandell, B. A. & Feldman, H. M. Tract profiles of white matter properties: automating fiber-tract quantification. *PLoS one* **7**, e49790 (2012).
22. Colby, J. B. *et al.* Along-tract statistics allow for enhanced tractography analysis. *NeuroImage* **59**, 3227–3242 (2012).

23. Schilling, K. G. *et al.* Prevalence of white matter pathways coming into a single white matter voxel orientation: The bottleneck issue in tractography. *Hum Brain Mapp* **43**, 1196–1213 (2022).
24. Zhang, H., Schneider, T., Wheeler-Kingshott, C. A. & Alexander, D. C. NODDI: practical in vivo neurite orientation dispersion and density imaging of the human brain. *NeuroImage* **61**, 1000–1016 (2012).
25. Ellerbrock, I. & Mohammadi, S. Four in vivo g-ratio-weighted imaging methods: Comparability and repeatability at the group level. *Hum. Brain Mapp.* **39**, 24–41 (2018).
26. Campbell, J. S. W. *et al.* Promise and pitfalls of g-ratio estimation with MRI. *NeuroImage* **182**, 80–96 (2018).
27. Mohammadi, S. *et al.* Whole-Brain In-vivo Measurements of the Axonal G-Ratio in a Group of 37 Healthy Volunteers. *Front. Neurosci.* **9**, 441 (2015).
28. Graf von Keyserlingk, D. & Schramm, U. Diameter of axons and thickness of myelin sheaths of the pyramidal tract fibres in the adult human medullary pyramid. *Anat. Anz.* **157**, 97–111 (1984).
29. Da Lu, W. *et al.* Mapping the aggregate g-ratio of white matter tracts using multi-modal MRI. *Imaging Neurosci.* **3** (2025).
30. Genz, A. & Bretz, F. *Computation of Multivariate Normal and t Probabilities* (Springer Berlin Heidelberg, 2009).

Reviewer #1 (Remarks to the Author):

I thank the the authors for the replies to the concerns that were raised. Multiple concerns resonated also in the comments/suggestions of other authors. Although some are resolved, I feel that major concerns remain.

We thank the reviewer for the thoughtful and constructive comments. Below, we address each point in detail and describe the corresponding revisions made.

The first one being related to the wording and statements that are used. In the abstract the authors write about "neocortical reorganization". These are words that do not fit the extend of changes that are observed in such works.

lines 77-79: DTI is not an imaging modality, it is a model that is applied to a MRI protocol, namely diffusion weighted MRI or Diffusion weighed imaging. None of the models or techniques that are proposed in the paper do provide direct biological specificity. DTI is limited in specificity as do the other techniques, but there is no total lack of mapping.

lines 85-93: Overstatement of NODDI's biological validity. NODDI parameters are not directly histologically validated in humans, and validation in animals is limited and not always consistent. FICVF does not uniquely represent neurite density. For example, it is also influenced by other factors such as extracellular volume, axon diameter distribution, model assumptions...

Calling it neurite density is an oversimplification..

We appreciate the reviewer's helpful comments. Several formulations in the manuscript have been revised to improve accuracy and avoid overstatement.

1. Use of the term "neocortical reorganization" in the Abstract

This wording implies a degree of large-scale structural change that may convey an exaggerated impression of the nature of changes. We have replaced it with a more neutral and appropriate description of the observed effects ("neocortical plasticity").

2. Statement that "DTI is not an imaging modality"

This is correct. Diffusion Tensor Imaging is a signal representation applied to diffusion-weighted MRI data [1,2], not an independent modality. We have revised the manuscript accordingly. We also agree that our wording regarding a "lack of specificity" was too categorical. We clarified that DTI provides relatively limited biological specificity, rather than implying a complete absence of meaningful microstructural information. The sentence in question now reads:

"However, conventional imaging approaches such as T1-weighted MRI or tensor-based representations of diffusion MRI data (DTI) provide only limited biological specificity, as the measured signals cannot be mapped unambiguously to underlying tissue microstructural properties." (for references, see manuscript)

3. NODDI's biological validity and neurite density

We appreciate this important point. Validation of diffusion MRI models of white matter microstructure is generally difficult [3]. Indeed, it is very challenging to validate NODDI parameters in humans, and animal validations are limited as well. The fixation of tissue inevitably leads to an alteration of tissue properties, for instance in terms of reduced extra-cellular in relation to intra-cellular space [2]. We therefore toned down

our original phrasing (“yields histologically validated information”) and now more cautiously describe the interpretational constraints of the model:

“For example, fitting the biophysical model Neurite Orientation Dispersion and Density Imaging (NODDI) to high-gradient diffusion MRI data provides estimates of local tissue properties, including the isotropic volume fraction (FISO), the intracellular volume fraction (FICVF), and the orientational coherence of the neurites (orientation dispersion index, ODI). Although these parameters cannot be interpreted as direct histological measurements, they have been shown to correlate with microstructural tissue properties ex vivo. Compared with conventional DTI, they provide improved specificity and serve as sensitive markers for detecting microstructural changes.” (for references, see manuscript)

At the same time, we highlight that NODDI and related “standard model” [1,2] approaches are among the more extensively studied diffusion-MRI-based microstructural models, with several ex vivo validation efforts [4–9], including studies on human tissue [4,5,7] and nonhuman primates [6].

Nonetheless, we basically agree with the reviewer that the “neurite density index” (NDI) might suggest a level of specificity that it cannot offer (e.g., [4,10]). We want to clarify that the term was not coined by us, but by the developers of the NODDI model themselves [11], and has since become common usage in the literature. Note that this terminological issue does not only apply to NODDI, but can also be found in a similar way in other biophysical models of diffusion (for instance, the “axonal water fraction” (AWF) in WMTI [12]).

Following the reviewer’s suggestion, we have revised terminology throughout the manuscript. In the Methods section, we now use a more neutral and precise description:

“[...] the intracellular volume fraction (FICVF, commonly referred to as neurite density index, NDI), quantifying the fraction of tissue volume occupied by neurites and other cell processes [...]”

Figure 1 is misleading in its current state. If the authors refer to “changes” on the left side of the figure, how could they indicate an increase or decrease in the outcome parameters. for example, myelin changes can be of different nature. not only an increase in thickness is possible.

We thank the reviewer for this helpful comment and agree that Figure 1 requires revision.

The sketches on the left side of the figure were intended to schematically illustrate microstructural plasticity processes as commonly depicted in the review literature, such as changes in axonal diameter and mesoscale organization, relative axonal myelination, or glial cell hypertrophy [13,14]. However, we agree that in its current form the figure may unintentionally suggest a specific directionality of change in MRI-derived microstructural measures.

As the reviewer correctly notes, this issue is particularly relevant for myelin. Changes in network efficiency can result from both increases and decreases in myelin, as long as these processes support adjustments in axonal conduction velocity and a precisely timed arrival of action potentials at key relay points within a network [15–17]. Moreover, myelin plasticity is multifaceted—not only involving new myelin formation and changes in sheath thickness, but also modulation of internode length, alterations at the nodes of Ranvier, or changes in the number of oligodendrocyte progenitor cells [13].

We acknowledge that capturing the full complexity of cellular plasticity mechanisms and their relationship to MRI-derived measures within a single figure is challenging. Nevertheless, we believe that a carefully revised figure can still substantially aid clarity and readability.

In response to the reviewer's concerns, we will revise Figure 1 as follows:

- Remove explicit directionality (i.e., avoid indicating increases or decreases), since both upward and downward changes in imaging measures may be functionally meaningful.
- Address the nuances of myelin plasticity in the figure legend rather than in the figure itself. Incorporating all aspects of myelin remodeling directly into the graphic would require substantial space and risk distracting from the main message. We therefore propose to keep the figure conceptually clear while elaborating on the complex biological underpinnings more thoroughly in the caption.

Revised figure and caption:

Figure 1. Schematic overview of proposed mechanisms of structural neuroplasticity in white matter (WM), as inferred from animal studies, and the quantitative MRI (qMRI) metrics that have been reported to be associated with these microstructural features. From top to bottom, the illustrated cellular and microstructural mechanisms include changes in axonal diameter and mesoscale neurite organization, myelination, and glial cell number and/or hypertrophy. The schematic depicts associations between these processes and MRI-derived measures without implying a specific direction of change. Although several additional MRI metrics (including some illustrated) are sensitive to alterations in axonal properties, myelin, and neuroglia, only those with the highest relative specificity are shown here to maintain conceptual clarity. As a schematic abstraction, the figure does not capture the full, multifaceted nature of myelin plasticity, which may involve changes in sheath thickness, internode length, node of Ranvier properties, or oligodendrocyte progenitor dynamics.

We trust that these revisions will resolve the concern and improve the interpretability of the figure.

Suggestion to use aggregate g-ratio throughout the manuscript, also in the abstract.

Implemented as suggested by the reviewer.

With respect to R1Q3, I appreciate the additional explanation. At the same time, these findings still require some reasoning. Why would the left and right CST "react" modality-specific. What could be a plausible reason for this. In the rebuttal it is not mentioned whether this is now added to the main manuscript.

In light of the demonstrated reliability of our imaging measures [18,19] and the behavioral relevance of the observed effects, we consider it unlikely that the reported hemispheric differences in CST plasticity simply reflect chance. At the same time, our data do not allow for a definitive, evidence-based explanation of why the left and right CST would exhibit modality-specific responses. We therefore provide plausible interpretations rooted in existing literature and known neurobiological principles.

Lateralized plasticity is a well-established phenomenon across learning paradigms [20] and has also been reported specifically in white-matter pathways involved in balance control [21–23]. Thus, hemispheric asymmetries per se are not unexpected. We speculate that the observed modality-specific CST responses reflect the fact that balance training may engage the left and right hemispheres differently, thereby recruiting distinct microstructural plasticity mechanisms. In our data, the *right CST* exhibited a pattern of changes that is consistent with increased relative myelination, with the aggregate *g*-ratio as the most important driver. In contrast, the more inferiorly located effect in the *left CST* is harder to interpret. An increase in PD could reflect transient cellular phenomena such as astrocytic swelling, whereas the concurrent R2* increase may point to iron-related processes—either glial alterations involving iron deposition or vascular mechanisms linked to heme-bound iron. Alternatively, the R2* change could be secondary to increased PD itself, since elevated water content enhances magnetic susceptibility and accelerates relaxation. Thus, while the biological interpretation—particularly of the left-hemispheric effect—remains uncertain, the observed pattern of results is compatible with known hemispheric asymmetries in structural plasticity.

These points are now explicitly discussed in the third paragraph of the Discussion section, where we present them as a speculative but biologically plausible interpretation of the modality-specific CST effects.

R1Q4: I thank the reviewers for pointing out that CC and SLF were included in their analyses. What I do not understand is why the negative findings / null findings in these tracts are not discussed as one would anticipate to see changes there based on previous literature.

In addition, I understand that a division in 100 segments based on the TractSeg framework is interesting from a practical point of view, but what about the scientific basis for this large number of segments, regardless of total length of the tract?

Finally, disregarding the start and end point could mean that we are missing interesting parts. These are the parts that reach close to the GM and therefore are potentially more prone to plasticity.

We thank the reviewer for these thoughtful comments and agree that they raise important conceptual and methodological issues. We have revised the manuscript accordingly and clarify our reasoning below.

1. Corpus callosum (CC) and superior longitudinal fasciculus (SLF).

Recent reviews suggest that balance training induces widespread structural adaptations across the brain rather than changes confined to a small number of isolated regions [24]. This implies that plasticity may, in principle, be expected in many white-matter tracts. When structural adaptations are widespread and likely

modest in magnitude, detecting statistically robust effects in any single tract becomes inherently challenging, particularly under conservative analytical criteria.

In the present study, plasticity was defined stringently, requiring not only training-related structural change but also a demonstrable relationship with behavioral improvement. This dual criterion was intentionally adopted to emphasize functional relevance, but it is more restrictive than the approaches used in many previous studies and reduces sensitivity to effects that may be present but behaviorally nonspecific.

Furthermore, as is common in neuroimaging research, the moderate sample size limits statistical power to reliably detect medium-sized effects in individual white-matter tracts [25,26]. Under these constraints, the absence of statistically significant findings in the CC and SLF should not be interpreted as evidence that these tracts are unaffected by balance training. Rather, the null results likely reflect limitations in detectability given the distributed nature of training-induced plasticity, the conservative definition of plasticity, and statistical power considerations. Accordingly, absence of evidence should not be taken as evidence of absence.

We have now added the following sentences to the Discussion:

“Beyond these, additional limitations relate to sample characteristics and statistical power. The moderate sample size limits sensitivity to detect tract-specific effects of modest magnitude. Balance training likely induces broadly distributed white-matter adaptations rather than focal changes, but stringent plasticity criteria as applied here reduce the probability of detecting modest tract-specific effects in any single tract.” (for references, see manuscript)

2. Scientific basis for subdivision into 100 segments.

With respect to the subdivision of tracts into 100 segments, we emphasize that this choice is not specific to TractSeg. Other commonly used tractometry frameworks, including pyAFQ [27] and DSI Studio [28], employ the same convention. To our knowledge, there is no explicit biological rationale for selecting exactly 100 segments, and any fixed subdivision necessarily arguably involves an element of arbitrariness. However, dividing tracts into a standardized number of nodes allows normalization across tract length and subjects, thereby facilitating anatomical correspondence and group-level comparisons. Notably, one could also take the opposite position and argue that 100 segments is rather coarse, especially when compared to voxel-based analyses (see R2Q1 and R2A1). From our perspective, however, 100 segments represent a pragmatic compromise between spatial granularity and statistical stability. We now explicitly discuss this issue in the Discussion section:

“Segment-level averaging, while potentially diluting highly localized effects, represents a practical compromise: it enhances power and reduces noise relative to voxel-wise analyses while retaining anatomical specificity. Pure voxel-wise approaches offer higher spatial resolution but face challenges such as misregistration, low signal-to-noise ratio, and severe multiple-comparison penalties.” (for references, see manuscript).

A systematic evaluation of how tract subdivision granularity influences sensitivity to localized plasticity constitutes a methodological question in its own right and was beyond the scope of the present study.

3. Exclusion of tract start and end regions.

We agree with the reviewer that tract endpoints—particularly regions near the gray–white matter boundary—may contain biologically interesting information and could, in principle, be sensitive to plastic changes. However, their exclusion in the present study was motivated by methodological considerations.

Specifically, diffusion estimates near cortical terminations are more susceptible to crossing fibers and partial volume effects at the gray–white matter boundary. For these reasons, Huber and colleagues [29] averaged only the middle 60% of each tract. Thus, while tract endpoints may be biologically relevant, current diffusion MRI tractometry does not yet allow these regions to be analyzed with sufficient reliability to support robust inference. Our decision to analyze the middle 80% of each tract therefore reflects a trade-off between capturing potentially informative regions and ensuring robust, comparable, and interpretable diffusion measures.

R1Q5: since also one of the other reviewers asked about this, I suggest to add this information to the text and not limited to a statement in the discussion. I thank the authors for performing these additional analyses.

As suggested by the reviewer, the results of this analysis are now provided in Supplementary Figure 1. In the Results section, we are now referencing to these results:

“In an additional sensitivity analysis, female participants (n = 3) were excluded, and the analyses were repeated in the remaining male participants. The results were largely consistent with the primary analyses, with some outcome measures showing slightly increased stability during the control period and/or more pronounced neuroplastic effects following training. Importantly, no qualitative changes in the pattern of results were observed. These findings suggest that inclusion of the female participants does not materially affect the robustness of our conclusions (Supplementary Figure 1).”

R1Q6: I thank the authors for adding these correlation plots. These however do not look convincing. Can the authors change the axis ranges to get a better view of the spread. The Y axis range is narrow, making the trends look steeper? It shows high scatter relative to trend lines which could indicate substantial individual variability, effect sizes may not be practically meaningful.

We thank the reviewer for this comment and the opportunity to clarify the presentation of the correlation plots.

Because the scatterplots are based on ranked values, their appearance differs from that of conventional scatterplots using raw data. All correlations were computed using Spearman’s rank correlation; accordingly, both the x- and y-axes display ranked values of the 24 participants (i.e., ranks 1–24). As a result, the axes necessarily share identical and relatively narrow ranges, and the apparent steepness of the trend lines reflects this rank-based scaling rather than a restricted y-axis or visual distortion.

Spearman’s correlation was chosen deliberately because it provides a conservative and outlier-robust estimate of monotonic associations [30], which is particularly appropriate given the sample size and inter-individual variability in the data. Regarding practical significance, the observed effect sizes can be interpreted directly from the correlation coefficients. Following Cohen’s guidelines [31], correlations in the range of $.30 < |r| < .50$ correspond to medium-sized effects, which we consider meaningful and realistic in the context of the present study.

References

1. Novikov, D. S., Fieremans, E., Jespersen, S. N. & Kiselev, V. G. Quantifying brain microstructure with diffusion MRI: Theory and parameter estimation. *NMR Biomed.* **32**, e3998 (2019).
2. Jelescu, I. O., Palombo, M., Bagnato, F. & Schilling, K. G. Challenges for biophysical modeling of microstructure. *J. Neurosci. Methods* **344**, 108861 (2020).
3. Jelescu, I. O. & Budde, M. D. Design and validation of diffusion MRI models of white matter. *Front. Phys.* **28** (2017).
4. Grussu, F. *et al.* Neurite dispersion: a new marker of multiple sclerosis spinal cord pathology? *Ann. Clin. Transl. Neurol.* **4**, 663–679 (2017).
5. Mollink, J. *et al.* Evaluating fibre orientation dispersion in white matter: Comparison of diffusion MRI, histology and polarized light imaging. *NeuroImage* **157**, 561–574 (2017).
6. Schilling, K. G. *et al.* Histological validation of diffusion MRI fiber orientation distributions and dispersion. *NeuroImage* **165**, 200–221 (2018).
7. Seppehrband, F. *et al.* Brain tissue compartment density estimated using diffusion-weighted MRI yields tissue parameters consistent with histology. *Hum. Brain Mapp.* **36**, 3687–3702 (2015).
8. Wang, N. *et al.* Neurite orientation dispersion and density imaging of mouse brain microstructure. *Brain Struct. Funct.* **224**, 1797–1813 (2019).
9. Gong, N.-J., Dibb, R., Pletnikov, M., Benner, E. & Liu, C. Imaging microstructure with diffusion and susceptibility MR: neuronal density correlation in Disrupted-in-Schizophrenia-1 mutant mice. *NMR Biomed.* **33**, e4365 (2020).
10. Fukutomi, H. *et al.* Neurite imaging reveals microstructural variations in human cerebral cortical gray matter. *NeuroImage* **182**, 488–499 (2018).
11. Kraguljac, N. V., Guerreri, M., Strickland, M. J. & Zhang, H. Neurite Orientation Dispersion and Density Imaging in Psychiatric Disorders: A Systematic Literature Review and a Technical Note. *Biol. Psychiatry Glob. Open Sci.* **3**, 10–21 (2023).
12. Fieremans, E., Jensen, J. H. & Helpert, J. A. White matter characterization with diffusional kurtosis imaging. *NeuroImage* **58**, 177–188 (2011).
13. Sampaio-Baptista, C. & Johansen-Berg, H. White Matter Plasticity in the Adult Brain. *Neuron* **96**, 1239–1251 (2017).
14. Zatorre, R. J., Fields, R. D. & Johansen-Berg, H. Plasticity in gray and white: neuroimaging changes in brain structure during learning. *Nat. Neurosci.* **15**, 528–536 (2012).
15. Pajevic, S., Plenz, D., Basser, P. J. & Fields, R. D. Oligodendrocyte-mediated myelin plasticity and its role in neural synchronization. *eLife* **12** (2023).
16. Kato, D. *et al.* Motor learning requires myelination to reduce asynchrony and spontaneity in neural activity. *Glia* **68**, 193–210 (2020).
17. Fields, R. D. A new mechanism of nervous system plasticity: activity-dependent myelination. *Nat. Rev. Neurosci.* **16**, 756–767 (2015).
18. Aye, N. *et al.* Test-retest reliability of multi-parametric maps (MPM) of brain microstructure. *NeuroImage* **256**, 119249 (2022).
19. Lehmann, N. *et al.* Longitudinal Reproducibility of Neurite Orientation Dispersion and Density Imaging (NODDI) Derived Metrics in the White Matter. *Neuroscience* **457**, 165–185 (2021).
20. Esteves, M., Ganz, E., Sousa, N. & Leite-Almeida, H. Asymmetrical Brain Plasticity: Physiology and Pathology. *Neuroscience* **454**, 3–14 (2021).

21. Im, S., Oh, J., Jun, S. Y., Chang, S.-Y. & Kim, Y. Evidence of lateralised white matter plasticity: A longitudinal study of balance performance in nonexpert healthy adults. *Eur. J. Neurosci.* **57**, 1789–1802 (2023).
22. Taubert, M. *et al.* Dynamic properties of human brain structure: learning-related changes in cortical areas and associated fiber connections. *J. Neurosci.* **30**, 11670–11677 (2010).
23. Lehmann, N., Villringer, A. & Taubert, M. Priming cardiovascular exercise improves complex motor skill learning by affecting the trajectory of learning-related brain plasticity. *Sci. Rep.* **12**, 1107 (2022).
24. Surgent, O. J., Dadalko, O. I., Pickett, K. A. & Travers, B. G. Balance and the brain: A review of structural brain correlates of postural balance and balance training in humans. *Gait Posture* **71**, 245–252 (2019).
25. Szucs, D. & Ioannidis, J. P. Sample size evolution in neuroimaging research: An evaluation of highly-cited studies (1990-2012) and of latest practices (2017-2018) in high-impact journals. *NeuroImage* **221**, 117164 (2020).
26. Szucs, D. & Ioannidis, J. P. A. Empirical assessment of published effect sizes and power in the recent cognitive neuroscience and psychology literature. *PLoS Biol.* **15**, e2000797 (2017).
27. Kruper, J. *et al.* A software ecosystem for brain tractometry processing, analysis, and insight. *PLoS Comput. Biol.* **21**, e1013323 (2025).
28. Yeh, F.-C. DSI Studio: an integrated tractography platform and fiber data hub for accelerating brain research. *Nat. Methods* **22**, 1617–1619 (2025).
29. Huber, E., Donnelly, P. M., Rokem, A. & Yeatman, J. D. Rapid and widespread white matter plasticity during an intensive reading intervention. *Nature communications* **9**, 2260 (2018).
30. de Winter, J. C. F., Gosling, S. D. & Potter, J. Comparing the Pearson and Spearman correlation coefficients across distributions and sample sizes: A tutorial using simulations and empirical data. *Psychol. Methods* **21**, 273–290 (2016).
31. Cohen, J. *Statistical power analysis for the behavioral sciences*. 2nd ed. (Erlbaum, 1988).